# WHAMM initiates autolysosome tubulation by promoting actin polymerization on autolysosomes

Anbang Dai [1], Li Yu [2] & Hong-Wei Wang [1]

WHAMM, a member of the Wiskott-Aldrich syndrome protein (WASP) family, is an actin nucleation promoting factor (NPF) that also associates with membranes and microtubules. Here we report that WHAMM is required for autophagic lysosome reformation (ALR). WHAMM knockout causes impairment of autolysosome tubulation, which results in accumulation of enlarged autolysosomes during prolonged starvation. Mechanistically, WHAMM is recruited to the autolysosome membrane through its specific interaction with $PI(4,5)P_2$. WHAMM then works as an NPF which promotes assembly of an actin scaffold on the surface of the autolysosome to promote autolysosome tubulation. Our study demonstrates an unexpected role of the actin scaffold in regulating autophagic lysosome reformation.

[1] Ministry of Education Key Laboratory of Protein Sciences, Tsinghua-Peking University Joint Center for Life Sciences, Beijing Advanced Innovation Center for Structural Biology, School of Life Science, Tsinghua University, 100084 Beijing, China. [2] State Key Laboratory of Membrane Biology, Tsinghua-Peking University Joint Center for Life Sciences, School of Life Science, Tsinghua University, 100084 Beijing, China. Correspondence and requests for materials should be addressed to L.Y. (email: liyulab@tsinghua.edu.cn) or to H.-W.W. (email: hongweiwang@tsinghua.edu.cn)

Autophagy is an evolutionarily conserved degradation pathway[1]. Deactivation of mTOR signaling upon a stimulus such as starvation initiates autophagy and triggers formation of autophagosomes. The autophagosomes then fuse with lysosomes and form autolysosomes for the degradation of their contents. After cargo degradation, the release of products such as amino acids re-activates mTOR and terminates autophagy[2,3]. The autolysosomes subsequently go through a process called autolysosome reformation (ALR), in which the lysosomal membrane components are recycled from autolysosomes to regenerate lysosomes[2].

Branched actin networks play important roles in shaping, organizing, and maintaining intracellular membrane systems[4,5]. It was recently recognized that actin play multiple roles in regulation of autophagy[6]. A few studies have demonstrated that actin can regulate autophagosome formation and autophagosome-lysosome fusion. For example, CapZ was reported to promote autophagosome shaping by regulation of branched actin network assembly in the isolation membrane[7]. Moreover, MyosinVI and cortactin have been shown to regulate autophagosome-lysosome fusion[8,9].

Branched actin networks are generated by a family of proteins known as Wiskott-Aldrich syndrome proteins (WASPs), which activate the Arp2/3 complex to promote actin nucleation for branched actin filament growth. These proteins with actin nucleation promoting activity are called NPFs (nucleation promoting factors). Each WASP family member has distinct roles in different sites and pathways within the cell. For example, N-WASP is known to be involved in Clathrin-mediated endocytosis[10]. Actin is polymerized by N-WASP towards the Clathrin-coated pit to enhance membrane deformation and push the pit inward, thus facilitating the budding of Clathrin-coated pits[11,12].

WHAMM is a member of the WASP family with a conserved C-terminal WCA domain that also exists in other WASP family members to activate the Arp2/3 complex in actin nucleation[13]. Beside its role as an NPF, WHAMM can also associate with membranes and microtubules. WHAMM was originally shown to function in ER-to-cis-Golgi transport[13]. More recently, WHAMM has been shown to be involved in autophagy, and is required for autophagosome biogenesis[14].

In the current work, we discover an unexpected role of WHAMM in autophagy. We find that, instead of influencing autophagosome biogenesis, WHAMM is mostly involved in autolysosome reformation (ALR). We further demonstrate that WHAMM promotes actin polymerization to initiate autolysosome tubulation during ALR. Finally, we find that WHAMM is recruited to autolysosomes by PI(4,5)P2, and the PI(4,5)P2-binding activity of WHAMM is essential for WHAMM to facilitate autolysosome tubulation. Our study reveals a new role of WHAMM and branched actin networks in regulation of autophagy.

## Results

**WHAMM is required for autolysosome reformation.** Recently, WHAMM was shown to co-localize with LC3 and to be required for autophagosome biogenesis. Knockdown of WHAMM caused reduced size and number of autophagosomes[14]. We confirmed that WHAMM indeed co-localizes with LC3 in Normal Rat Kidney (NRK) cells (Fig. 1a). In order to better understand the role of WHAMM in autophagy, we knocked out WHAMM in NRK cells using the CRISPR-Cas9 system. We then expressed GFP-LC3 in both WT and WHAMM-KO cells and monitored autophagosome formation. To our surprise, after 2 h of starvation, the average number of LC3 puncta in WHAMM-KO cells appeared to be similar to WT (Fig. 1b, c). Additionally, the

pattern of LC3 turn-over was similar in WT and WHAMM-KO cells (Fig. 1d). Using transmission electron microscopy (TEM), we observed that autophagosomes in WHAMM-KO cells showed normal morphology, as in WT cells (Fig. 1e). These data show that the lack of WHAMM in the cells does not affect autophagosome maturation.

Next, we examined whether WHAMM may be involved in other stages of autophagy. It has been well established that once an autophagosome becomes mature, it fuses with lysosomes to initiate the digestion of its cargos[3]. This newly formed compartment is called an autolysosome, as it is positive for both the autophagosome marker LC3 and the lysosome marker LAMP1[2]. We starved NRK cells stably expressing mCherry-WHAMM and stained them with antibodies against LC3 and LAMP1. After starvation for 4 h, WHAMM showed enhanced co-localization with LAMP1, and these WHAMM-LAMP1 puncta were also LC3 positive, which suggests that they are autolysosomes (Fig. 1f, g). This finding raises the possibility that WHAMM may actually be engaged in later stages of autophagy.

After cargo degradation, autolysosomes go through a recycling process named autolysosome reformation (ALR), so as to restore the number of lysosomes within the cell[2]. One of the most distinguishing features of ALR is the tubulation of autolysosomes, driven by PI(4,5)P2, Clathrin and KIF5B kinesin[15,16]. Small LAMP1-positive compartments called proto-lysosomes are generated by scission from these reformation tubules[15]. Eventually, proto-lysosomes mature into lysosomes to replenish the cellular pool of lysosomes. In order to test if WHAMM is involved in ALR, we co-transfected CFP-LC3 and LAMP1-mCherry into both WT and WHAMM-KO cells and induced starvation at 18 h post transfection. We found that at 8 h of starvation, reformation tubules were hardly observed in WHAMM-KO cells, while they were clearly visible in wild-type cells (Fig. 2a). Statistical analysis showed that tubulation was almost completely abolished in the absence of WHAMM (Fig. 2b). This phenotype indicates that WHAMM is needed for autolysosome tubulation. On the other hand, previous studies have shown that blocking ALR can cause accumulation of enlarged autolysosomes after prolonged starvation[15]. Indeed, we also observed much larger LC3- and LAMP1-positive autolysosomes in WHAMM-KO cells 12-h post-starvation (Fig. 2c). When we measured the size of these autolysosomes, we saw a significant change of their distribution towards larger sizes (Fig. 2d). This observation was verified by TEM analysis of both WT and WHAMM-KO cells. TEM micrographs showed that WHAMM-KO cells have many more autolysosomes than WT cells (Fig. 2e, f). Correlative light and electron microscopy (CLEM) further confirmed these enlarged LC3- and LAMP13-positive structure are autolysosomes (Fig. 2g). Together, these data suggest that WHAMM knockout causes a defect in the ALR process, and WHAMM may play a crucial role in autolysosome tubulation.

**WHAMM's function as an NPF is indispensable for ALR.** To test the relationship between the NPF activity of WHAMM and ALR, we generated a truncated version of WHAMM, which lacks the WCA domain (1-630), and a point mutated version (W807A), which was reported to lose its ability to promote actin polymerization in vitro[13]. To verify their NPF activity, we purified MBP-tagged recombinant proteins for each mutant and full-length (FL) WHAMM. These proteins were firstly applied to a pyrene actin polymerization assay. We found that FL WHAMM, but not the 1-630 or W807A mutants, significantly enhanced actin polymerization (Fig. 3a). This result confirmed that 1-630 and W807A lack the ability to promote actin nucleation. We then

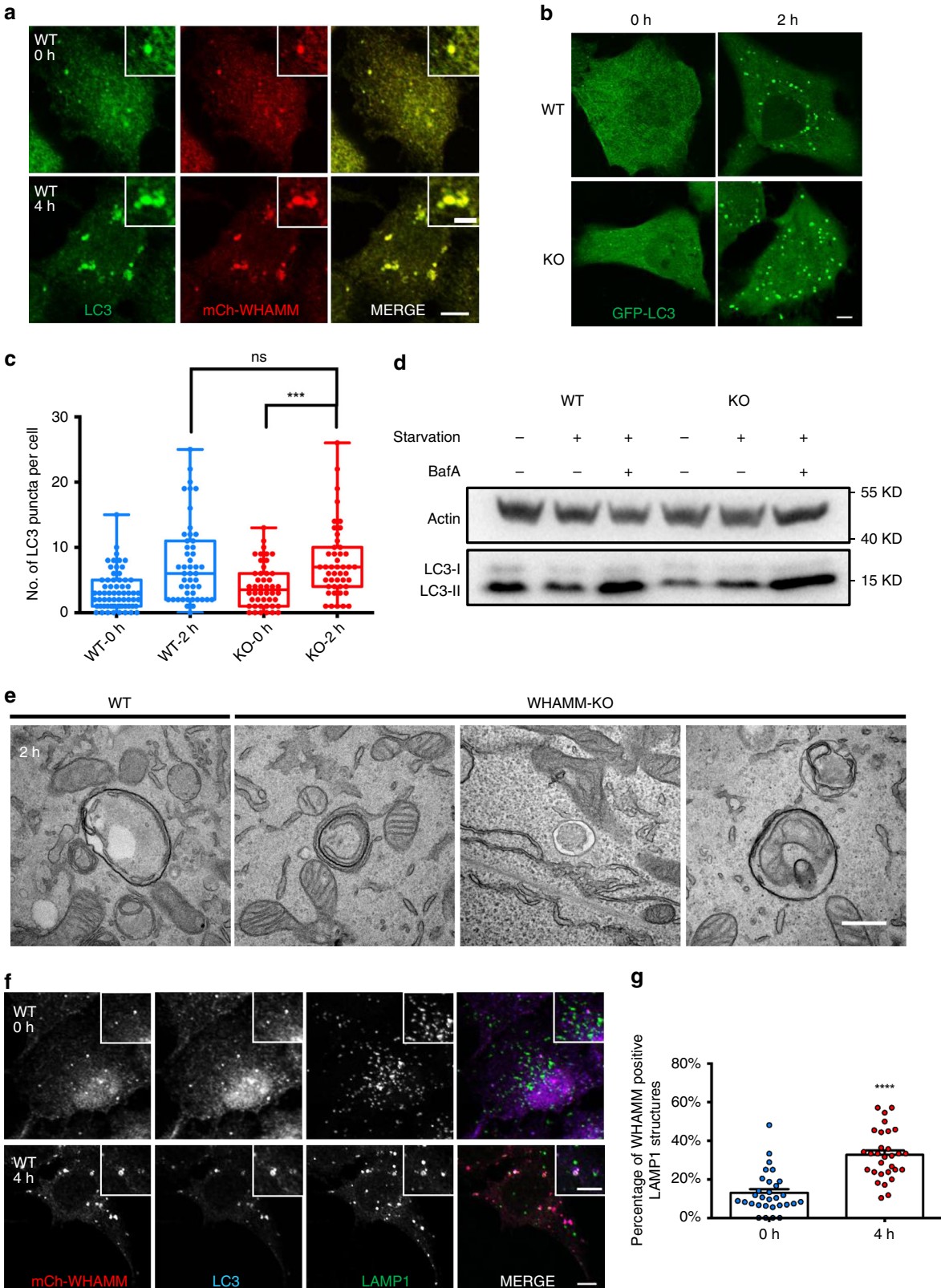

transfected mCherry-tagged versions of these three constructs into WHAMM-KO cells that stably express LAMP1-YFP, to see if they can rescue the defective autolysosome tubulation. Eighteen hours after transfection, cells were starved for 8 h and examined under a confocal microscope. Autolysosome tubulation was well restored in cells expressing FL WHAMM, but was hardly visible in cells expressing 1-630 and W807A (Fig. 3b and Supplementary Fig. 2a, b). Moreover, at 12 h post-starvation, LAMP1-positive compartments returned to their normal size in cells expressing FL WHAMM, but remained enlarged in cells expressing either 1-630 or W807A (Supplementary Fig. 2c, d). These data suggest that the NPF function of WHAMM is essential for ALR.

**Fig. 1** WHAMM knockout cells show normal levels of autophagosome maturation. **a** NRK cells stably expressing mCherry-WHAMM were starved for 0 or 4 h, then fixed and stained with antibodies against LC3 (scale bar, 5 μm). **b** CRISPR-Cas9-mediated gene knockout was used to generate a WHAMM knockout NRK cell line. Knockout verification and off-target analysis are shown in Supplementary Fig. 1. Both WT and WHAMM-KO cells stably expressing GFP-LC3 were starved for 2 h and observed using confocal microscopy (scale bar, 5 μm). **c** Cells in (**b**) were quantified for LC3 puncta. $n = 65$ (WT-0h), 46 (KO-0h), 48 (WT-2h), and 46 (KO-2h) cells were measured from two independent experiments. Box extends from 25th to 75th percentiles. Middle line indicates median. Whiskers represent min to max with all points shown. Two-tailed $t$ test; ***$p < 0.001$; ns, not significant. **d** Autophagic flux was monitored by treating both WT and KO cells with Bafilomycin-A1 (BafA) after starvation. LC3 turnover was examined by western blot. **e** Representative TEM images of autophagosomes from both WT and KO cells after 2 h of starvation (scale bar, 500 nm). **f** NRK cells stably expressing mCherry-WHAMM were starved for 0 or 4 h, then fixed and stained with antibodies against LC3 and LAMP1 (scale bar, 5 μm). **g** Cells in **f** were measured for WHAMM-positive LAMP1 structures and quantified. $n = 30$ cells from two independent experiments. Error bars indicate SEM. Two-tailed $t$-test; ****$p < 0.0001$. Source data are provided as a Source Data file

**WHAMM generates branched actin network on autolysosomes**. We therefore speculated that WHAMM may regulate autolysosome tubulation by generating branched actin networks on autolysosomes. To test this hypothesis, we treated WT cells stably expressing CFP-LC3 and LAMP1-mCherry with CK666, a cell-permeable drug that inhibits Arp2/3 activity but does not affect NPF binding[17]. Notably, it has been reported before that the branched actin network may be required for autophagosome shaping and fusion with lysosomes[7,9]. To rule out any effect of CK666 on the early stages of autophagy, we added the drug at 4 h after autophagy was induced, when most of the early events had already taken place (Fig. 3c). In the control group treated with DMSO, the level of tubulation appeared to be normal at 8 h post-starvation (Fig. 3d, e), and the size of autolysosomes was also restored at 12 h (Supplementary Fig. 2e, f). In contrast, cells treated with CK666 showed a typical defective ALR phenotype with much less tubulation at 8 h post-starvation (Fig. 3d, e) and enlarged autolysosomes at 12 h, very similar to the WHAMM-KO NRK cells (Supplementary Fig. 2e, f). Moreover, autolysosome tubulation was restored when CK666 was washed out, which indicates that CK666 specifically blocks autolysosome tubulation (Supplementary Fig. 3, Supplementary Movie 3). These data further indicate that autolysosome tubulation is dependent on formation of branched actin networks and raise the possibility that WHAMM may facilitate autolysosome tubulation by promoting branched actin networks on autolysosomes.

To test this hypothesis, we labeled cells with LifeAct and LAMP1 together. We found that actin formed puncta on autolysosomes after 8 h of starvation (Fig. 3f, g, Supplementary Movie 4) and WHAMM co-localized with these actin puncta on autolysosomes (Supplementary Fig. 4a). To investigate the nature of these actin puncta, we labeled cells with mCherry-Arp3, a subunit of the Arp2/3 complex[18], and mCherry-Cortactin, a protein that stablizes the branched actin network during autophagosome-lysosome fusion[9]. We found that WHAMM co-localized with these branched actin markers on autolysosomes (Fig. 4d, e), which also indicated that the actin puncta on autolysosomes contained branched actin network. In WHAMM-KO cells, the percentage of these actin-positive autolysosomes was significantly reduced (Fig. 3g, h), suggesting that WHAMM directs actin network formation on autolysosomes. Moreover, FL WHAMM, but not the 1-630 or W807A mutants, rescued the recruitment of actin to autolysosomes (Fig. 3h, Supplementary Fig. 4b), emphasizing the crucial roles of WHAMM's NPF activity for the branched actin network formation on autolysosomes.

By taking time-lapse images, we were able to observe that the actin puncta displayed a locally enriched pattern through time, specifically at the budding site on the main bodies of autolysosomes (Fig. 3f and Supplementary Movie 4). In consistence with this observation, we found that WHAMM also localized on the main body of autolysosomes (Fig. 4a, Supplementry Movie 1) and the base of a newly formed reformation tubule (Fig. 4b,

Supplementary Movie 2). Moreover, although a few WHAMM puncta were occasionally observed on the tubules, they were spaced far apart and not densely packed (Fig. 4c).

In summary, these data suggest that WHAMM regulates autolysosome tubulation by promoting the formation of branched actin networks on autolysosomes. The fact that the branched actin network and WHAMM are localized on the main body of autolysosomes and at the base of reformation tubules implies that WHAMM may facilitate the initiation of autolysosome tubulation by generating a pushing force through the branched actin network.

**WHAMM binds PI(4,5)P$_2$ in vitro**. WHAMM contains two domains not found in the other WASPs, a unique N-terminal WHAMM membrane-binding domain (WMD, 1-260) and a central coiled-coil region (CC, 260-570)[13,19]. From protein-lipid overlay assays, WHAMM was initially discovered to be able to bind phosphoinositides through its WMD domain[13]. However, it is generally accepted that protein-lipid overlay assays are prone to artifacts. To better characterize the binding between phospholipids and WHAMM, we generated 400-nm-sized liposomes made of 1,2-dioleoyl-sn-glycero-3-phosphocholine (DOPC) and 1,2-dio-leoyl-sn-glycero-3-phosphoethanolamine (DOPE). These liposomes were doped with different types of phosphatidylinositol phosphates (PIPs) or simply phosphatidylinositol (PI) at an equal concentration of 15%. We used a liposome flotation assay to examine the interaction of WHAMM with these liposomes (Fig. 5a). In brief, we mixed 5 μg FL WHAMM with 25 μl 1 mM liposomes containing different PIPs. After centrifugation, liposomes migrated to the top of the gradient along with bound proteins. Each fraction was then gently collected and examined by SDS-PAGE. We found that WHAMM showed a prominent interaction with PI(4,5)P$_2$; we also observed a weak interaction with PI(3,5)P$_2$ and PI(3,4,5)P$_3$. It is worth noting that we did not see any noticeable interaction of WHAMM with PI or other PIPs (Fig. 5b, c).

To further study the membrane association preference of WHAMM, we made liposomes of different sizes containing 15% PI(4,5)P$_2$. We used WHAMM with a C-terminal GFP-tag in the flotation assay, and monitored the GFP-fluorescence in the top fraction after the centrifugation. Interestingly, we found that more WHAMM associated with larger liposomes than with smaller ones (Fig. 5d). This suggests that WHAMM may prefer to bind membranes with smaller curvature. Moreover, sedimentation assays using liposomes containing different concentrations of PI(4,5)P$_2$ indicated that WHAMM had a clear preference to bind to liposomes containing the higher concentration of PI(4,5)P$_2$ (Fig. 5e). We also found that adding 10% cholesterol, which can enhance the local PI(4,5)P$_2$ concentration by clustering PI(4,5)P$_2$[20], enabled binding of WHAMM to liposomes containing a physiological level of PI(4,5)P$_2$ (Fig. 5f). Since PI(4,5)P$_2$ is enriched on bud-like structures on the autolysosome surface, these data suggest that WHAMM may be recruited to autolysosomes by PI(4,5)P$_2$.

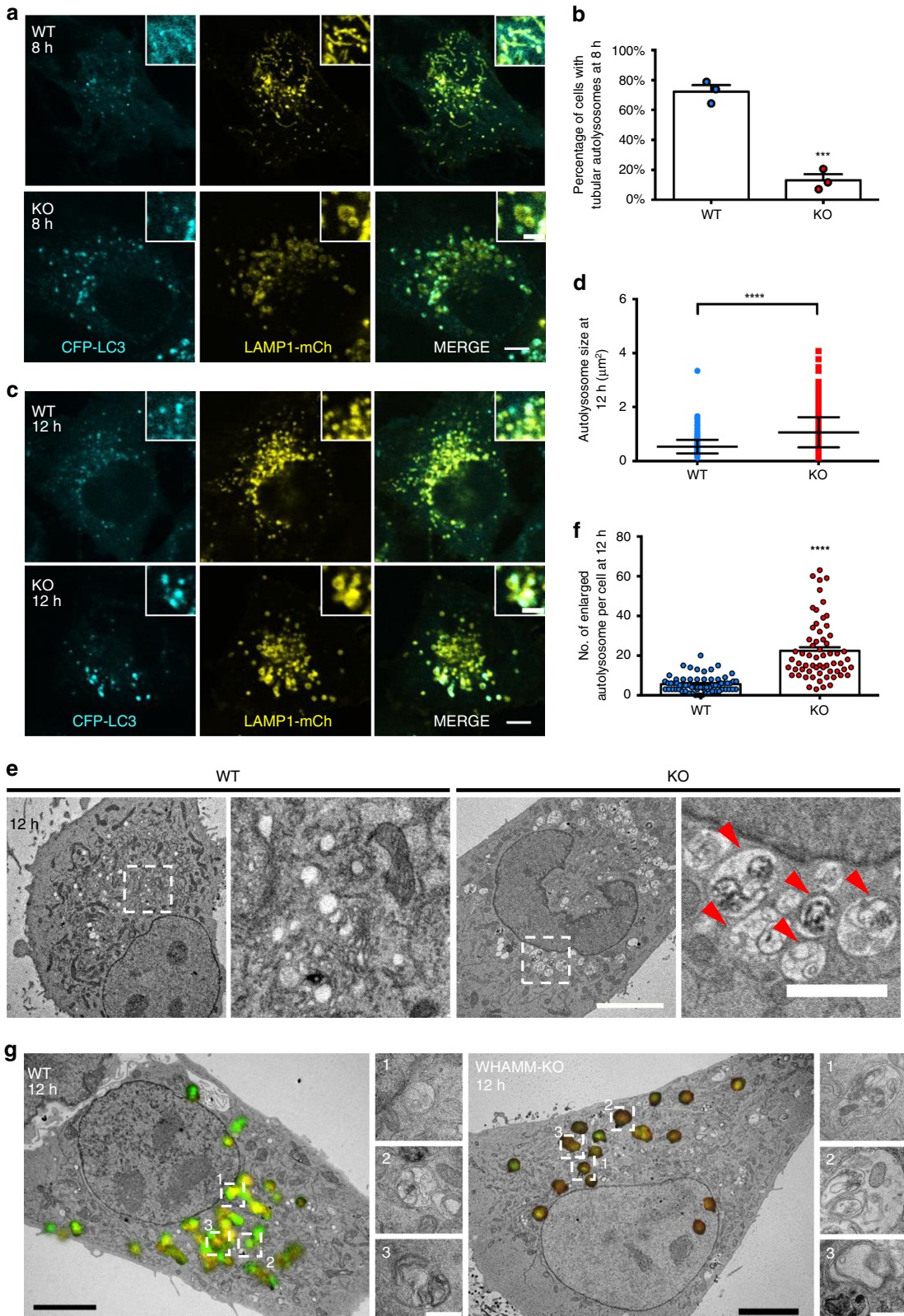

**WHAMM binds PI(4,5)P₂ through two amphipathic helices**. The current consensus is that WHAMM binds to phospholipids through its N-terminal 260 amino acids, which are known as the WMD domain[13]. Surprisingly, in our in vitro binding assays, we found that when we deleted the WMD domain, the remaining part of WHAMM was still able to bind to PI(4,5)P₂-containing liposomes (Fig. 6a).

We sought to find the structural elements of WHAMM that mediate its interaction with PI(4,5)P₂-containing membranes. There are several well-defined modules that facilitate protein

**Fig. 2** WHAMM localizes on autolysosomes and is required for autolysosome reformation. **a** Both WT and KO cells were transfected with CFP-LC3 and LAMP1-mCherry. 18 h after transfection, cells were starved for 8 h, then observed using confocal microscopy. LAMP1-mCherry was pseudo-colored to yellow (scale bar, main Fig. 5 μm; upper right panel 2 μm). **b** Cells in **a** were assessed for tubular autolysosomes and quantified. A total of 103 (WT) and 120 (KO) cells were examined in $n = 3$ independent experiments. Error bars indicate SEM. Two-tailed $t$-test; ***$p < 0.001$. **c** Cells were treated as in **a** but further starved to 12 h and observed using confocal microscopy. LAMP1-mCherry was pseudo-colored to yellow (scale bar, main Fig. 5 μm; upper right panel 2 μm). **d** The size of autolysosomes was measured in cells in **c**. $n = 799$ (WT) and 824 (KO) autolysosomes were randomly selected and measured in three independent experiments. The size of individual autolysosomes is plotted in each column. Error bars indicate SD. The middle horizontal line indicates the mean. Two-tailed $t$-test; ****$p < 0.0001$. **e** Representative TEM images of both WT and KO cells after 12 h of starvation. Enlarged autolysosomes are indicated with red arrowheads (scale bar, main micrograph 5 μm; inset 2 μm). **f** Number of enlarged autolysosomes per cell from **e** was quantified. $n = 66$ (WT) and 62 (KO) cells were analyzed in two independent experiments. Error bars indicate SEM. Two-tailed $t$-test; ****$p < 0.0001$. **g** Both WT and WHAMM-KO NRK cells were transfected with CFP-LC3 (green) and LAMP-mCherry (red). In all, 18 h post transfection, cells were starved for 12 h and then fixed with 4% paraformaldehyde (PFA). Fluorescence images were taken with a confocal microscope. Cells were then further fixed with 2.5% glutaraldehyde (GA) and prepared for EM analysis. Representative LAMP1- and LC3-positive structures are enclosed in dashed squares and shown in the right panels. A detailed description can be found in the Methods section. (Scale bar, main Fig. 5 μm; right panels 500 nm). Source data are provided as a Source Data file

interactions with PI(4,5)P$_2$, such as pleckstrin (PH)[21] and FERM domains[22]. However, WHAMM does not harbor any sequences homologous to these modules. In addition, unlike its relative N-WASP, which binds PI(4,5)P$_2$ through a polybasic region[23], WHAMM does not contain a dense region of basic or aromatic residues. We closely inspected the N-terminal region of WHAMM for potential phospholipid binding motifs and found two interesting helices: helix 1 (188–208) and helix 2 (319–339), which are located separately within residues 1–260 and 260–340 (Fig. 6b, c). These two helices are predicted by secondary structure algorithms and are conserved among species (Fig. 6c and Supplementary Fig. 5a). Using helical wheel representations, we found that both helices have similar amphipathicity and residue arrangements, with basic residues on both sides of the hydrophobic face, which could potentially bind membranes comprising lipids with negatively charged head groups (Fig. 6d). Helices with such properties have also been found in other membrane-binding proteins such as ATG3[24] and BAR domain-containing proteins[25], and the helices are responsible for the interaction of the proteins with negatively charged lipids. Taking these features together, we suspected that these two regions might be responsible for the interaction of WHAMM with membranes containing PI(4,5)P$_2$.

To verify our hypothesis on helix 1 and helix 2's property to bind PI(4,5)P$_2$, we generated two constructs, 1-260 and 310-809, which harbor helix 1 and helix 2 respectively. Next, we introduced two individual point mutations (L190D, L201K) into 1-260 to disrupt the hydrophobic face of helix 1; similarly, we introduced one point mutation (A321D) into 310-809 to disrupt the hydrophobic face of helix 2 (Fig. 6d). We purified these mutant proteins and tested their ability to bind PI(4,5)P$_2$ using the flotation assay. Our results showed that when the hydrophobic faces of helix 1 and helix 2 were disrupted, they lost their ability to bind PI(4,5)P$_2$ (Fig. 6e, f). These results confirmed that helix 1 and helix 2 bind PI(4,5)P$_2$ with their hydrophobic face.

We further tested if mutations in both helices are sufficient to prevent FL WHAMM from binding PI(4,5)P$_2$. We created a dual mutant by introducing the L201D mutation into helix 1 and the A321D mutation into helix 2 to disrupt both helices simultaneously. We found that the ability to bind to PI(4,5)P$_2$-containing liposomes was completely abolished in the L201D-A321D mutant (Fig. 6e, f). Thus, WHAMM binds PI(4,5)P$_2$ solely through these two amphipathic helices.

**WHAMM is recruited to autolysosomes through PI(4,5)P$_2$.** PI(4,5)P$_2$ is present on autolysosomes and is a key player in ALR[15]. PI(4,5)P$_2$ on autolysosomes is mainly generated by PIP5K1B[15].

To test whether WHAMM is recruited to autolysosomes through PI(4,5)P$_2$, we first examined whether WHAMM is co-localized with the PI(4,5)P$_2$. Beside its use as a probe for PI(4,5)P$_2$ on plasma membranes, PLCδ-PH has also been used to label the intracellular pool of PI(4,5)P$_2$[26]. Indeed, we found that WHAMM co-localized with, or was in juxtaposition with, PLCδ-PH on autolysosomes (Fig. 7a). Next, we tested if WHAMM is recruited to autolysosomes through binding to PI(4,5)P$_2$ in vivo. We transfected cells with mCherry-tagged wild-type and L201D-A321D WHAMM plasmids, and found that the co-localization between LAMP1 and mutant WHAMM was markedly reduced (Fig. 7b, c). Thus, WHAMM is recruited to autolysosomes through binding to PI(4,5)P$_2$.

Next, we tested whether the PI(4,5)P$_2$-binding capacity of WHAMM is required for ALR. We transfected WT WHAMM and the L201D-A321D mutant into WHAMM-KO cells stably expressing LAMP1-YFP and observed the change in ALR within these cells after starvation (Fig. 7d, e). At 8 h post-starvation, we found that WT WHAMM rescued the tubulation phenotype, while the L201D-A321D mutant failed to rescue the autolysosome tubulation (Fig. 7d, e). Putting these data together, we concluded that WHAMM is recruited to autolysosomes through PI(4,5)P$_2$, and the recruitment of WHAMM to autolysosomes is required for autolysosome tubulation.

## Discussion

In this manuscript, we illustrated the role of WHAMM in the regulation of ALR. We found that WHAMM is recruited to autolysosomes during ALR. We further demonstrated that WHAMM can bind to PI(4,5)P$_2$, and WHAMM is recruited to autolysosomes through binding to PI(4,5)P$_2$. Once recruited onto the autolysosome surface, WHAMM promotes formation of branched actin networks, which facilitate the tubulation of autolysosomes (Fig. 8).

In contrast to another report[14], we did not find that WHAMM influenced autophagosome biogenesis. There are multiple possible reasons for this difference. First, we used a different cell line to conduct our experiments. Second, instead of using RNA interference, we used the CRISPR-Cas9 system to generate a WHAMM knockout cell line. Furthermore, instead of relying on transient transfection, we established a cell line with stable expression of WHAMM to study the localization of WHAMM during autophagy. All these factors may contribute to the different results obtained by us and the other group.

ALR is governed by complicated molecular machinery. Previous studies demonstrated that budding of tubulation sites is governed by Clathrin and its associated proteins, and the

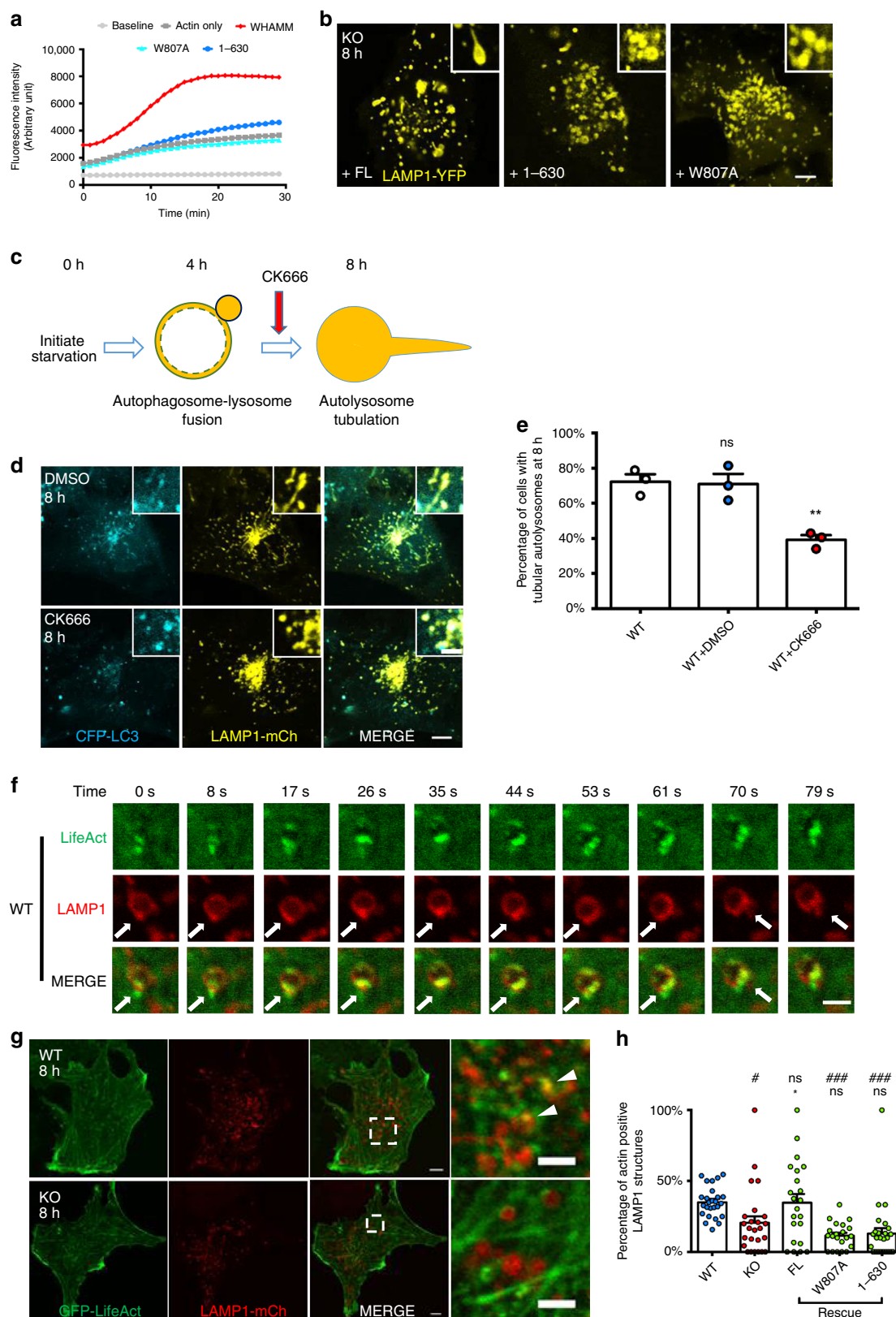

extension of tubules is driven by kinesin 1, which is recruited to budding sites through direct interaction with PI(4,5)P$_2$[15,16]. In the present study, we identified that the assembly of WHAMM-mediated branched actin networks on the autolysosome surface is an important step for initiation of ALR. At this moment, we still do not know how branched actin networks can facilitate tubulation; however, we did notice the striking similarity between initiation of Clathrin-mediated endocytosis and ALR. During endocytosis, the branched actin networks assemble around the Clathrin-coated pits, and the maturation of endocytic

**Fig. 3** WHAMM's actin nucleation promoting activity is required for ALR. **a** A pyrene actin polymerization assay was performed using purified WHAMM FL or different NPF-defective mutants. **b** Different constructs expressing WHAMM FL or NPF-defective mutants were transfected into WHAMM-KO cells stably expressing LAMP1-YFP. In all, 18 h post transfection, cells were starved for 8 h, then observed by confocal microscopy (scale bar, main micrograph 5 μm, upper right panel 2 μm). **c** A schematic drawing of the assay to test the effect of CK666 treatment on autolysosome tubulation. Drug was added 4 h after the onset of starvation to avoid its effects during the early events of autophagy. **d** WT NRK cells stably expressing CFP-LC3 and LAMP1-mCherry were starved for 4 h, then 100 μM CK666 or DMSO were added to the starvation medium. Cells were starved for a further 4 h (8 h in total), then observed with confocal microscopy. LAMP1-mCherry was pseudo-colored to yellow (scale bar, main micrograph 5 μm, upper right panel 2 μm). **e** Cells in **d** were quantified for tubular autolysosomes at 8 h. A total of 103 (WT), 120 (DMSO), and 158 (CK666) cells were analyzed in $n = 3$ independent experiments. Error bars indicate SEM. One-way ANOVA followed with Holm-Sidak's multiple comparisons test; \*\*$p < 0.01$; ns, not significant. **f** Time-lapse images of WT NRK cells expressing GFP-LifeAct and LAMP1-mCherry at 8 h post-starvation. Actin puncta on the autolysosome surface are indicated by the arrows (scale bar, 2 μm). **g** WT and WHAMM-KO cells were co-transfected with GFP-LifeAct and LAMP1-mCherry. 18 h post transfection, cells were starved for 8 h, then observed using confocal microscopy. Arrowheads indicate co-localized actin on autolysosomes (scale bar, main micrograph, 5 μm; inset, 2 μm). **h** Cells in **g** and (Supplementary Fig. 4b) were assessed for actin-positive LAMP1 structures. $n = 26$ (WT), 25 (KO), 22 (FL), 26 (1-630), and 21 (W807A) cells from two independent experiments. Error bars indicate SEM. One-way ANOVA followed with Holm-Sidak's multiple comparisons test. Compared with WT: \#\#\#$p < 0.001$; \#$p < 0.05$; ns, not significant. Compared with KO: \*$p < 0.05$; ns, not significant. Source data are provided as a Source Data file

vesicles is dependent on actin and its NPF N-WASP[27,28]. Moreover, the N-WASP is recruited to PI(4,5)P$_2$, which is enriched in Clathrin-coated pits[29,30]. The assembly of branched actin networks has been proposed to generate a mechanical force which propels the Clathrin-coated pit inward[31]. Once matured, the endocytic vesicle is then carried further inward by kinesin[32]. Similarly, during ALR, Clathrin mediates the formation of bud-like structures on autolysosomes, in which PI(4,5)P$_2$ is enriched, and recruitment of WHAMM by PI(4,5)P$_2$ promotes the assembly of branched actin networks on autolysosomes to facilitate the tubulation of autolysosomes. We speculate that similar to their role in Clathrin-mediated endocytosis, the branched actin networks may also provide a mechanical force, which coordinates with kinesin to provide the mechanical force required for tubulation (Fig. 8).

Endosomes undergo extensive tubulation during cargo recycling. Endosome tubulation is dependent on the NPF WASH, which localizes at microdomain on the surface of endosomes. Knockout of WASH results in enlarged endosomes with no tubules[33], which is very similar to the effect of WHAMM knockout on autolysosomes. Interestingly, there is also a discrepancy between the WASH siRNA knockdown and knockout phenotypes. In WASH knockdown cells, tubulation is not blocked; instead, endosomes shown extensive tubulation[34].

Lysosomes tubulation has been observed in LPS-exposed macrophages and dendritic cells[35]. Though induced by different stimuli, the similarities between ALR and lysosome tubulation in macrophages and dendritic cells are striking. Both are dependent on microtubules and driven by microtubule based motors[16,35–37], are governed by mTOR, and are regulated by Rab7[2]. Given so many similarities between the two processes, we speculate that they share the same regulatory pathway. It would be interesting to test whether the branched actin networks and NPFs also plays similar roles in lysosome tubulation as in ALR.

WHAMM has a specific microtubule-binding ability via its microtubule-binding motif[38] and can interact with microtubules in a cooperative way[19]. Therefore, it would not be surprising if WHAMM coordinates with kinesin to bring the autolysosomes in proximity to microtubules and help kinesin to pull the membranes along microtubules to form into tube-like shapes (Fig. 8). It remains a mystery why WHAMM mostly localizes to the ALR membranous region but not to the plasma membrane, given that both membranes have a high PI(4,5)P$_2$ content. WHAMM may interact with specific proteins that are related to the ALR process. Future studies may reveal more details of the mechanism by which WHAMM acts during the process of autophagic lysosome reformation.

## Methods

**Constructs**. mCherry-WHAMM and GFP-WHAMM were generated by cloning human WHAMM (1–809) into pmCherry-C1 and pEGFP-C1, respectively. Mutations and truncation constructs were generated using PCR-based mutagenesis. mCherry-Arp3 and mCherry-Cortactin were generated by cloning human Arp3 and rat Cortactin into pmCherry-C1, respectively. Other constructs were a kind gift from other members of the Yu Lab and were reported in previous studies[7,15,16]. All primers used in this study was listed in the Supplementary Information.

**Cell culture and transfection**. NRK cells were obtained from the American Type Culture Collection (ATCC) and cultured in DMEM (Hyclone, SH30022.01) supplemented with 10% Fetal Bovine Serum (Cellmax, SA112.01) under 5% CO$_2$ at 37 °C. Starvation was induced by adding DMEM (Life Technologies, 11960) after two washes in PBS. Cells were transfected with a total of 2 μg DNA by Amaxa nucleofection using solution T and program X-001. Stable cell lines were selected using 200 μg/ml Geneticin (Gibco, 10131-035). Positive single colonies were collected and expanded before use in subsequent assays.

**CRISPR-Cas9 knockout cell line**. To generate the WHAMM knockout NRK cell line, PX459(pSpCas9(BB)-2A-Puro) vector containing a guide sequence (5′-CACAGTCTAAGGGTGTGCGG-3′) was transfected into NRK cells. 24 h post transfection, cells were treated with 1 μg/ml puromycin (Beyotime, ST551) for 5 days. Single colonies were collected and cultured in 24-well plates. Each colony was expanded, then total genomic DNA was extracted and the target site was amplified using PCR. The PCR product was then ligated into a T-vector and transformed into DH5α competent cells. Single colonies were collected and sent for sequencing to verify that the modification resulted in successful knockout.

**Antibodies and reagents**. Primary antibodies were as follows: Anti-LC3 (MBL, PM036; Dilution ratio: 1:100 for immunofluorescence microscopy, 1:2000 for western blot.), Anti-LAMP1 (Enzo, ADI-VAM-EN001-F; Dilution ratio: 1:500 for immunofluorescence microscopy.), Anti-β Actin (Huaxingbio, HX1831; Dilution ratio: 1:5000 for western blot.). Secondary antibodies for immunofluorescence microscopy were from Invitrogen as follows: Goat anti-Rabbit IgG (H + L) Alexa Fluor 488: A-11008, Goat anti-Mouse IgG (H + L) Alexa Fluor 488: A-11001, Goat anti-Rabbit IgG (H + L) Alexa Fluor 647: A-21244, Goat anti-Mouse IgG (H + L) Alexa Fluor 647, A-21235). These antibodies were used at 1:500 dilution ratio. Secondary antibodies for western blot were from Huaxingbio as follows: HRP-Goat anti-Mouse IgG(H + L) (HX2032), HRP-Goat anti-Rabbit IgG (H + L) (HX2031). These antibodies were used at 1:5000 dilution ratio. Bafilomycin-A1 was purchased from Cell Signaling Technology (CST 54645S). Protease inhibitor cocktail tablets was purchased from Roche (04693132001).

**Immunofluorescence microscopy**. Cells were grown on coverslips in 24-well plates. Cells were washed with PBS and fixed using 4% paraformaldehyde (PFA) for 10 min at room temperature (RT), then permeabilized and blocked in 10% FBS in PBS containing 0.1% saponin for 1 h. Fixed cells were stained with primary antibody overnight at 4 °C, washed with PBS, and then stained with secondary antibody for 1 hour at RT and washed with PBS. Coverslips were mounted on slides and fixed using nail polish. Images were acquired using an Olympus FV-1200 confocal microscope or Nikon A1Rsi microscope.

**Live-cell imaging**. Cells were seeded on Lab-Tek chambered coverglasses (NUNC) 24 h before imaging. Cells were maintained at 37 °C with 5% CO$_2$ in an LCI chamber (LCI) during imaging. Images were acquired using an Olympus FV-1200 or Nikon A1Rsi confocal microscope. All fluorescence images were processed using FV10-ASW 3.1 software (Applied Precision) or Nikon NIS-elements, respectively.

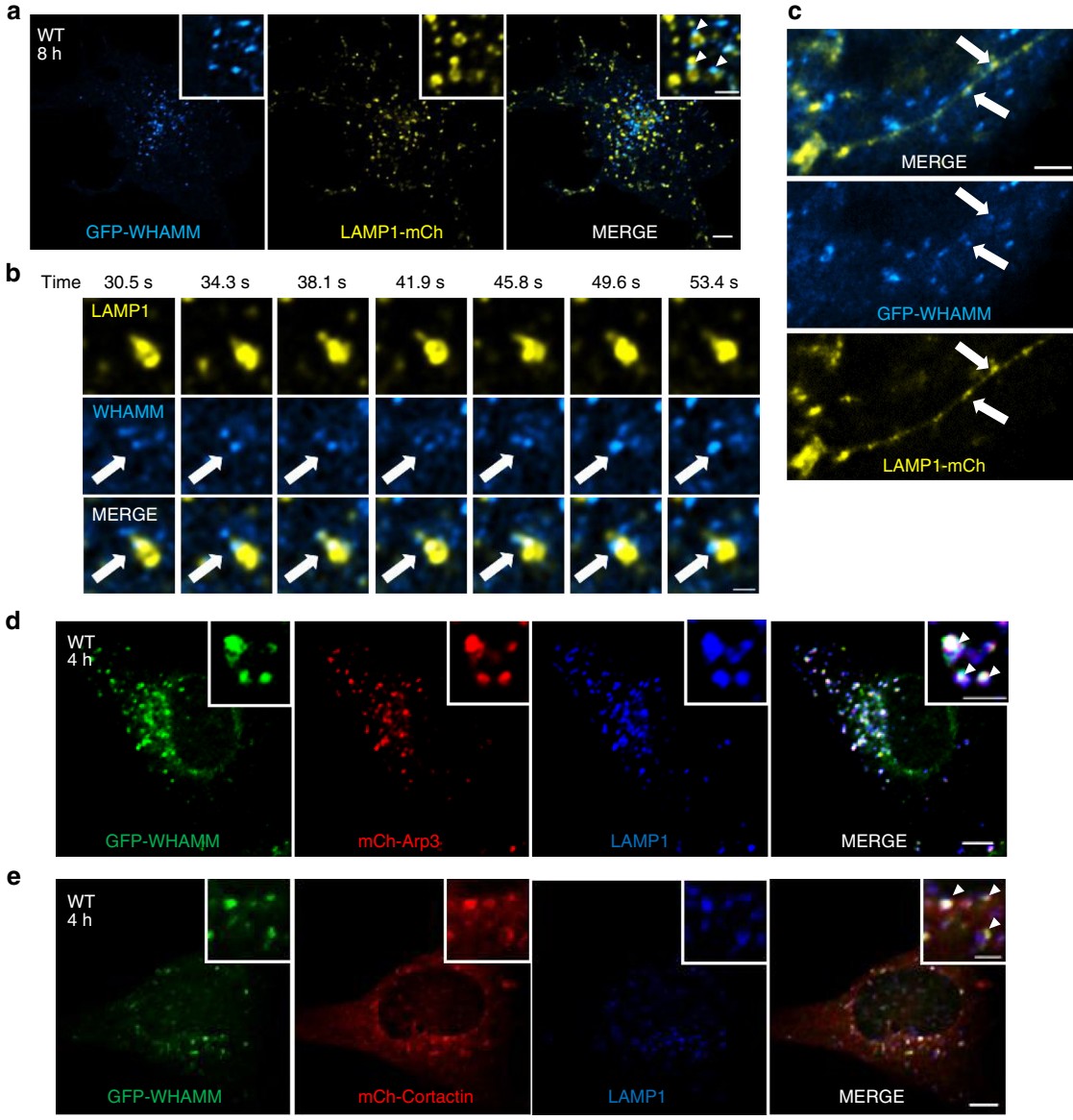

**Fig. 4** WHAMM generates branched actin network on autolysosomes. **a** WT NRK cells stably expressing GFP-WHAMM were transfected with LAMP1-mCherry. 18 h post transfection, cells were starved for 8 h and observed using live-cell imaging. GFP-WHAMM was pseudo-colored to cyan and LAMP1-mCherry was pseudo-colored to yellow. Arrowheads indicate WHAMM puncta on the surface of autolysosomes (scale bar, main Fig. 5 μm; upper right panel 2 μm). **b** Time-lapse images were taken of cells in **a**. Arrows indicate WHAMM puncta at the neck of a newly formed reformation tubule on an autolysosome (scale bar, 2 μm). **c** A snapshot of an elongated reformation tubule was extracted from cells in **a**. Arrows indicate WHAMM puncta on the reformation tubule. (Scale bar, 5 μm). **d**, **e** WT NRK cells stably expressing GFP-WHAMM were transfected with mCherry-Arp3 and mCherry-Cortactin, respectively. In all, 18 h post transfection, cells were starved for 4 h and then fixed and stained with antibody against LAMP1. Arrowheads indicate co-localization of WHAMM-Arp3 or WHAMM-Cortactin with autolysosomes (scale bar, main Fig. 5 μm; upper right panel 2 μm). All images in this figure were deconvolved using Nikon's built-in software (NIS-elements)

**Correlative confocal and electron microscopy**. Cells were seeded on a grid dish (Cellvis, D35-14-1.5GI) and initially fixed in 4% paraformaldehyde (PFA), then observed by confocal microscopy. After the confocal micrographs were taken, cells were further fixed in 2.5% glutaraldehyde (GA) for 2 h at RT. Cells were then dehydrated with a graded ethanol series (50%, 70%, 90%, 95%, and 100%) for 2 min each. Samples were infiltrated with and embedded in SPON12 resin, which was polymerized for 48 h at 60 °C. 70 nm-thick ultrathin sections were cut using a diamond knife, and then picked up with Formvar-coated copper grids (100 mesh). The sections were double stained with uranyl acetate and lead citrate. After air drying, cells were observed using a Hitachi, H-7650B transmission electron microscope at an acceleration voltage of 80 kV.

**Protein expression and purification**. WHAMM and its derivative proteins were expressed and purified from the HEK293-F suspension cell line. Cells were cultured under 37 °C and 8% $CO_2$. When the culture reached a density of 1.2–1.6 million cells per milliliter, cells were transfected with plasmids encoding respective proteins using PEI (poly-ethylenimine). 48–72 h post transfection, cells were harvested and lysed in ice-cold lysis buffer (20 mM Tris, pH 8.0, 250 mM NaCl, 100 mM KCl, 1 mM DTT, 1 mM EDTA, 5% Glycerol and protease inhibitor cocktails). Cell membrane was solubilized using 1% NP-40 for 1 h while rotating under 4 °C. Cell lysate was then centrifuged at 14,203 × $g$ using Type 45-Ti rotor at 4 °C for 1 hour. The supernatant was incubated with Amylose beads (NEB) for 2 h and bound proteins were eluted using 10 mM Maltose in lysis buffer. Target proteins were further purified using gel filtration column (Superdex 200, GE Healthcare).

**Liposome preparation**. All lipids were purchased from Avanti Polar Lipids, as follows: DOPC (850375), DOPE (850725), PI (840042), PI3P (850150), PI4P (840045), PI5P (850152), PI(3,4)P2 (850153), PI(3,5)P2 (850154), PI(4,5)P2 (840046), and PI(3,4,5)P3 (850156). To generate small unilamellar vesicles, 75% DOPC and 25% DOPE were mixed in a round-bottom glass tube and dried under

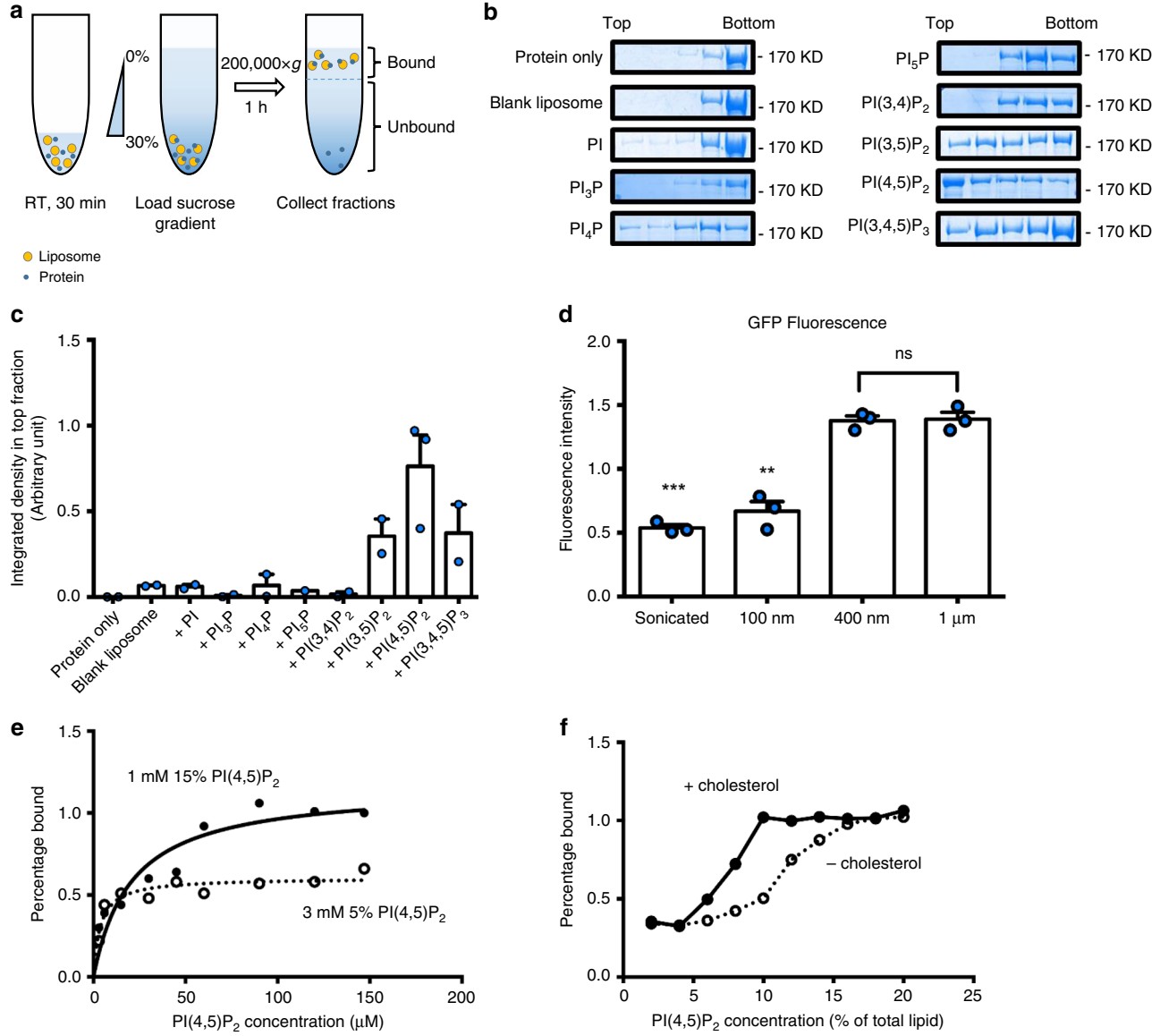

**Fig. 5** WHAMM binds to PI(4,5)P$_2$ in vitro. **a** A schematic representation of the liposome flotation assay. **b** Purified full-length WHAMM was incubated with DOPC-DOPE blank liposomes or liposomes doped with 15% of different PIPs as indicated, and then applied to the flotation assay. Each fraction was collected carefully and further analyzed by SDS-PAGE followed with Coomassie staining. **c** Band density in the top fraction from **b** and repeat experiments was measured using ImageJ. Each experiment was repeated two times except for PI$_5$P (once) and PI(4,5)P$_2$ (three times). Error bars indicate SEM. **d** GFP-tagged WHAMM was incubated with different sized liposomes containing 15% PI(4,5)P$_2$ and then applied to the flotation assay. The top fraction was collected and monitored by a fluorescence spectrophotometer to detect the GFP-fluorescence intensity. $n = 3$ independent experiments. Error bars indicate SEM. Two-tailed $t$-test; ***$p < 0.001$; **$p < 0.01$; ns, not significant. **e** Full-length WHAMM was incubated with liposomes containing a constant concentration of PI(4,5)P$_2$ but varying in density. The mixture was centrifuged down and the liposome pellet was subjected to SDS-PAGE. Band density was measured using ImageJ and normalized with input. **f** Liposomes containing increasing proportions of PI(4,5)P$_2$ with or without 10% cholesterol were incubated with full-length WHAMM. The mixture was then centrifuged down and the liposome pellet was subjected to SDS-PAGE. Band density was measured and normalized with input using ImageJ. Source data are provided as a Source Data file

an argon stream. Addition of phosphoinositide was counteracted by reducing the level of DOPC by an equal amount. The lipid film was further dried under vacuum overnight in order to remove remaining chloroform solvents. Liposome binding buffer (20 mM HEPES pH 7.4, 150 mM NaCl, 1 mM DTT) was added to the lipid film to achieve 1 mM total lipid concentration. After 10 cycles of freezing (liquid nitrogen, 1 min) and thawing (30 °C water bath, 3 min), liposomes were passed 31 times through polycarbonate films with a specific pore size. Sonicated liposomes were generated using a water bath sonicator.

**Liposome flotation and pelleting assay.** For the flotation assay, 5 μg of proteins were incubated with 25 μl of 1 mM liposomes for 30 min at room temperature (RT). The reaction mixture was then gently and thoroughly mixed with an equal

volume of liposome binding buffer containing 60% sucrose and transferred to the bottom of a centrifuge tube. A sucrose gradient was made from liposome binding buffer and constructed by consecutively overlaying 50 μl 25%, 50 μl 20%, and finally 50 μl 0% sucrose on top of the reaction solution. After ultracentrifugation using a TLS-55 rotor at 200,000 g for 1 h at 4 °C, fractions were carefully collected from the top and loaded on gels for SDS-PAGE. Band density was measured using ImageJ. The fluorescence intensity of GFP (488 nm) in the top fraction was measured using a fluorescence spectrophotometer (Hitachi, F4500). For the liposome pelleting assay, 2 μg of protein was mixed with increasing amount of liposomes, or the same amount of liposomes but with different PI(4,5)P$_2$ concentrations for 30 min at RT in a 50-μl total reaction volume. Then the reaction mix was centrifuged using a TLA-100 rotor at 24,441 × g for 30 min at 4 °C. The supernatant

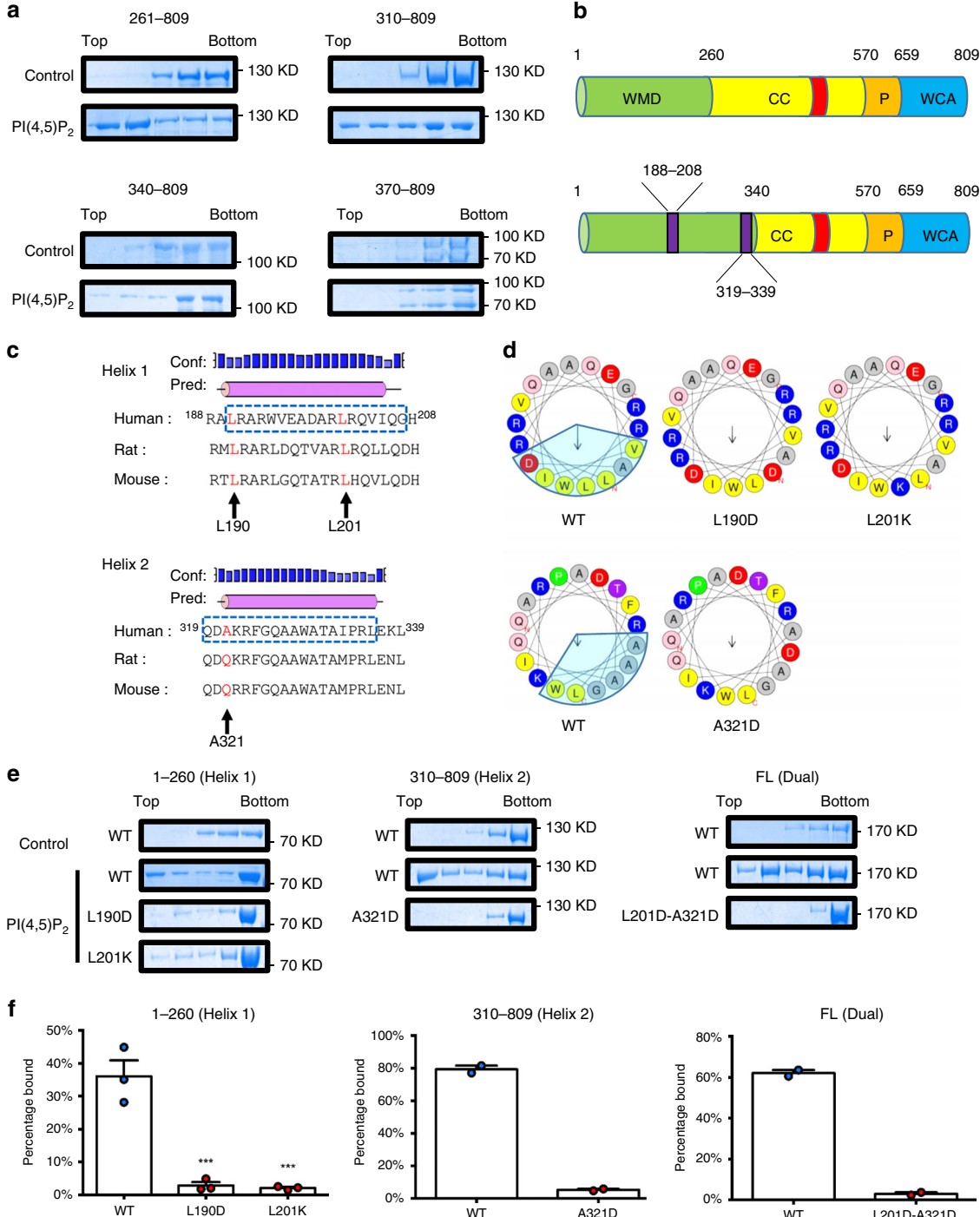

**Fig. 6** WHAMM binds to PI(4,5)P$_2$ through two conserved amphipathic helixes. **a** Flotation assays to identify the membrane-binding regions in WHAMM beyond the originally identified WMD (1-260) domain. Liposomes with only DOPC-DOPE were used as the control. **b** Comparison of the domain structure of WHAMM based on a previous model (top) and proposed in current work (bottom). Two regions suspected to be responsible for the membrane interaction of WHAMM are highlighted in purple. **c** Sequence alignment of helix 1 and helix 2, along with their secondary structure prediction[39]. The indicated point mutations (arrows) were introduced at the sites highlighted in red. The dashed box shows the region used to generate the helical wheel in the next panel. **d** Corresponding helical wheels of helix 1 and helix 2 generated using HELIQUEST[40]. The hydrophobic face is highlighted as a light blue sector. The black arrows indicate the direction of hydrophobic moments. The mutations that disrupt the hydrophobic face are also shown. **e** FL WHAMM and fragments containing helix 1 (1-260) or helix 2 (310-809) with and without the indicated mutations were purified and incubated with liposomes containing DOPC-DOPE only or with 15% PI(4,5)P$_2$ and applied to the flotation assay. Each fraction was collected carefully, then proteins were separated by SDS-PAGE and visualized using Coomassie blue. **f** Band density in the top fraction from each experiment in **e** was measured and normalized with total input using ImageJ. The proportion of protein bound to liposomes was compared between the mutants and WT from two or three independent experiments. Error bars indicate SEM. For 1-260 ($n = 3$), results were compared using one-way ANOVA followed with Holm-Sidak's multiple comparisons test. ***$p < 0.001$. For 310-809 ($n = 2$) and FL ($n = 2$). Source data are provided as a Source Data file

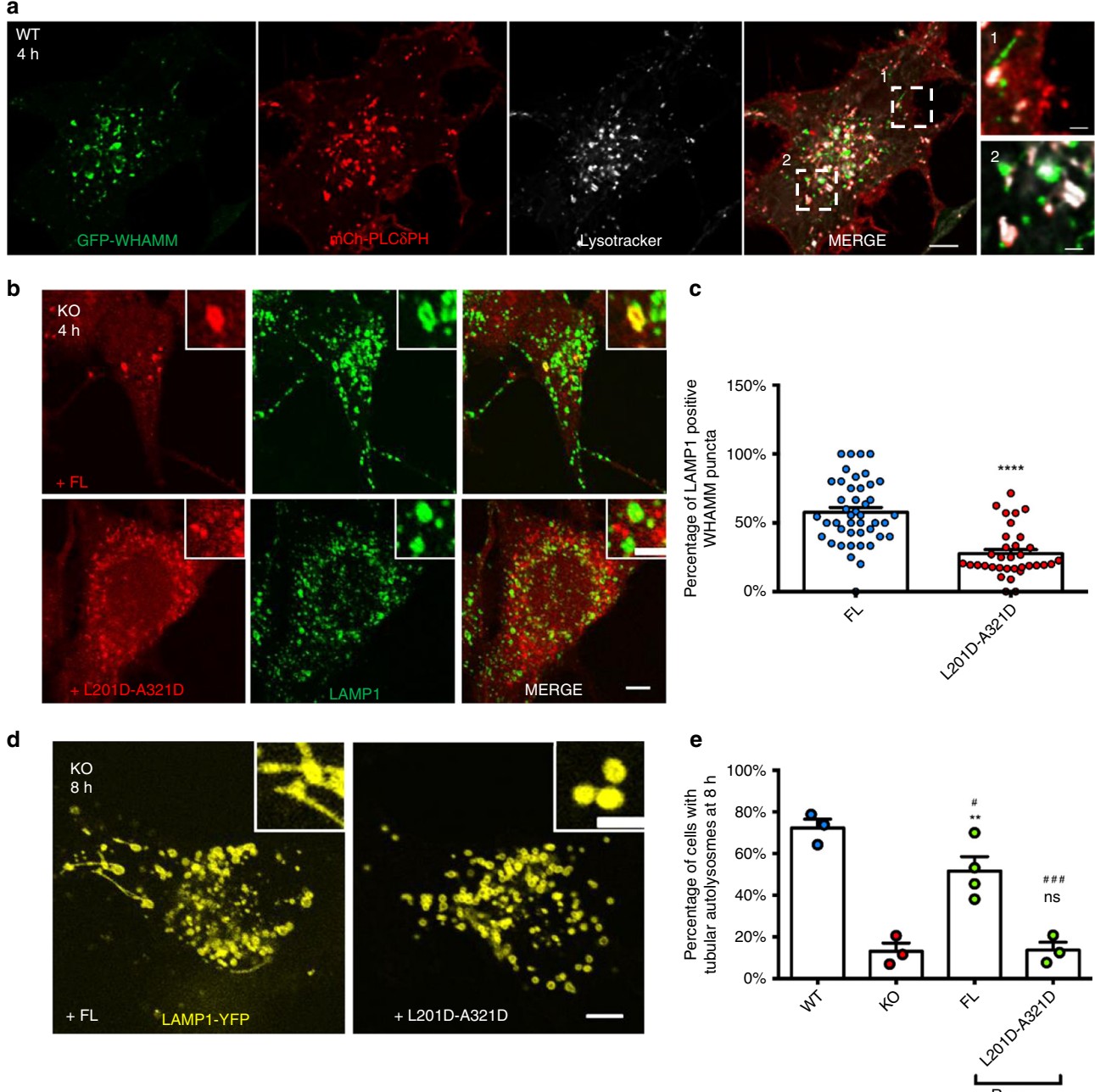

**Fig. 7** WHAMM is recruited to autolysosomes through PI(4,5)P$_2$. **a** WT NRK cells stably expressing GFP-WHAMM were transfected with mCherry-PLCδ-PH. 18 h post transfection, cells were starved for 4 h in the presence of 100 nM Lysotracker Deep Red and observed using live-cell imaging. Dashed squares showed WHAMM's localization correlates to Plasma membrane (1) and autolysosomes (2). (scale bar, main Fig. 5 μm; right panel 1 μm) These images were deconvolved using Nikon's built-in software (NIS-elements). **b** Constructs expressing FL WHAMM with or without the double mutation (L201D-A321D) were transfected into WHAMM-KO cells. After 18 h of transfection, cells were starved for 4 h, then fixed and stained with antibody against LAMP1 (scale bar, main micrograph 5 μm; upper right panel 2 μm). **c** Cells in **b** were assessed for WHAMM-positive LAMP1 structures. n = 44 (FL) and 34 (L201D-A321D) cells from two independent experiments. Error bars indicate SEM. Two-tailed t-test; ****p < 0.0001. **d** WHAMM-KO cells stably expressing LAMP1-YFP were transfected with constructs containing FL WHAMM with or without the double mutation (L201D-A321D). In all, 18 h post transfection, cells were starved for 8 h, then observed by confocal microscopy (scale bar, main micrograph 5 μm; upper right panel 2 μm). **e** Cells in **d** were quantified for tubular autolysosomes at 8 h. In total, 103 (WT), 120 (KO), and 50 (L201D-A321D) cells were examined in n = 3 independent experiments; 125 (FL) cells was analyzed in n = 4 independent experiments. Error bars indicate SEM. One-way ANOVA followed with Holm–Sidak's multiple comparisons test. Compared with KO: **p < 0.01; ns, not significant. Compared with WT: ###p < 0.001; #p < 0.05. Source data are provided as a Source Data file

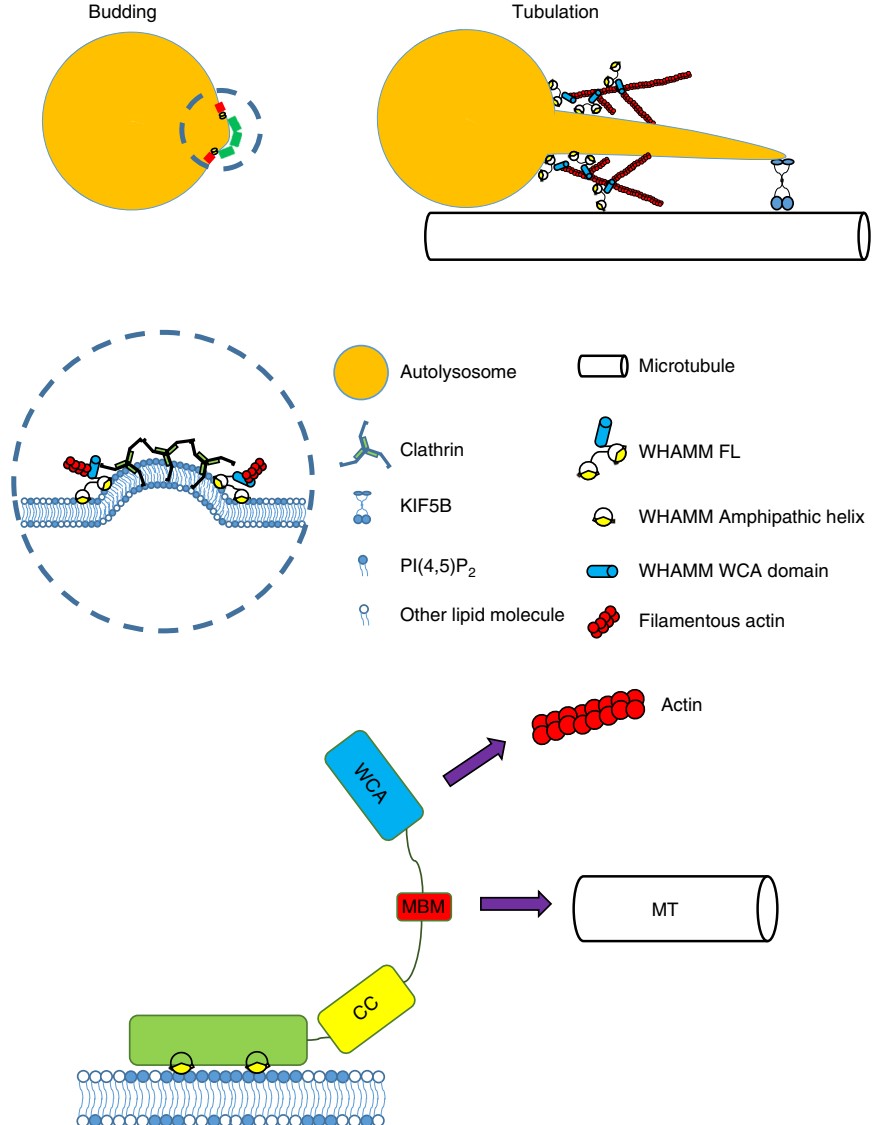

**Fig. 8** WHAMM promotes actin polymerization during autolysosome reformation. A schematic drawing showing the role of WHAMM during Clathrin-mediated autolysosome reformation. In brief, WHAMM binds to locally enriched PI(4,5)P$_2$ near Clathrin buds and generates branched actin networks to facilitate budding and tubulation of the autolysosome

was collected and the pellet was re-suspended with 1x SDS sample buffer. The total pellet suspension was subjected to SDS-PAGE and the band density was measured with ImageJ using the total input protein band as the reference. The percentage of binding was plotted and fitted with the One-site binding equation using Prism 6. All uncropped gel images were shown as Supplementary Fig. 9 and were also provided in Source Data file.

**Identification of autolysosomes and tubular structures**. Tubular autolysosomes are LAMP1-positive tubular structures extending from LAMP1-LC3 positive vesicular autolysosomes. This feature allows us to distinguish between autolysosomes and lysosomes when measuring the size of autolysosomes. The size of autolysosomes was measured using Image by encircling the autolysosomes using the oval selection tool. The area was then measured accordingly.

**Statistics**. All statistical analysis was performed using Prism 6. Details for each analysis can be found in the figure legends.

## Data availability

The data that support the findings of this study are available from the corresponding authors upon reasonable request. Source data for Fig. 1c, d, g; Fig. 2b, d, f; Fig. 3a, e, h; Fig. 5b–f; Fig. 6a, e, f; Fig. 7c, e; Supplementary Fig. 2b, d, f; Supplementary Fig. 6b; Supplementary Fig. 7b are provided in Source Data file.

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

## Acknowledgements

We thank Dr D.X. Sun, Dr W.Q. Du, and C. Jin for technical assistance, and Dr Y. Li for help with TEM sample preparation. We thank Dr D.C. Zhang and everyone in the Yu Lab and Wang Lab for comments. This work was supported by grant (2016YFA0501100 to H.W.) from the Ministry of Science and Technology of China, grant (Z161100000116034 to H.W.) from the Beijing Municipal Science & Technology Commission.

## Author contributions

A.D., L.Y. and H.W. conceived the idea. A.D. designed and performed the experiments and analyzed data under the supervision of L.Y. and H.W.; A.D., L.Y., and H.W. wrote the manuscript.

## Additional information

**Competing interests:** The authors declare no competing interests.

