## [Peer Review File · Nature Communications]

Reviewers' comments:

Reviewer #1 (Remarks to the Author):

This study focuses on the role of WHAMM in lysosome reformation from autolysosomes. The authors demonstrate that the protein is required for not autophagosome formation but for lysosome reformation in contrast to the previous work from the other group. They show that WHAMM promotes actin network formation on autolysosomes through its actin nucleation ability, which is involved in autolysosome tubulation, an initial step of lysosome reformation. They further found that WHAMM is recruited to autolysosomes by PI(4,5)P₂ and its binding to this PIP is essential for lysosome reformation. They locate the PIP binding site in WHAMM.

Overall this paper is logical and well written. The data is timely and provides new mechanistic insights into lysosome reformation from autolysosomes, which is an important step in autophagy but still not well understood. I expect that this work will be not only of interest to the autophagy community but also of broad interest to researchers in membrane biology field, providing a conceptual advance for understanding how cells manage organelle dynamics.

Needed improvements:

1. To confirm that the structures in TEM images in Fig 2e are identical to dots labeled with both LC3 and LAMP1 in Fig 2c, the authors should perform CLEM (Correlative light and electron microscopy) method or immuno EM. In Fig 2e, vesicles in the KO cells include a lot of membranes, suggesting that degradation inside autolysosomes is suppressed. The authors should examine a possibility that WHAMM is also involved in autolysosomal degradation by autophagy flux assay.
2. In WHAMM-KO cells, the actin-positive autolysosomes are reduced (Fig 3h, i). The authors should show that FL but not 1-630 and W807A mutants of WHAMM rescue the phenotype.
3. As the authors discuss, since the plasma membrane contains a lot of PI(4,5)P₂, there must be other factor(s) that determine the specific binding of WHAMM to autolysosomes. Although the authors mention that future studies may reveal this point, it is desirable that the present paper includes identification of a candidate of binding partner of WHAMM on autolysosomes other than PI(4,5)P₂.

Reviewer #2 (Remarks to the Author):

In this manuscript, Dai and colleagues report a novel function of WHAMM in autophagic lysosome reformation, which is a late step of autophagy. To this end, they have generated WHAMM KO in NRK cells and they perform rescue with different constructs of WHAMM. In particular, they show using specific mutants that Arp2/3 activation and PIP₂ binding are critical for WHAMM to perform its late function in autophagy.

Overall this manuscript is nice and clear. It is well written and the reader has a sense of logical progression until the end. The techniques are appropriate and the results are convincing. In terms of originality, there were already reports that WHAMM was involved in autophagy, but only in an early step, autophagosome biogenesis. The implication of WHAMM in late autophagy as described here is completely novel and well documented in the manuscript. A conundrum though is that the early function was revealed using siRNAs, whereas the late function described here is revealed using CRISPR-Cas9 generated KO, so incomplete knock-down is not the reason...

I believe that this work should be published provided that minor improvements are included. I also suggest experiments that may extend the manuscript. As such, they are not absolute requirements, but to my sense would increase the number of reasons of citing this manuscript when it is going to be published.

Required improvements

1 The actin mutants of WHAMM were previously described; the PIP2 mutants are new and are clearly a major advance of the manuscript. The PIP2 mutants are inactive as expected. However, I believe that a clear colocalization of WT WHAMM with a PIP2 probe is lacking. To most investigators, PIP2 is mostly located at the plasma membrane. Here it would also be useful to see the relative staining of autolysosomes and plasma membrane. If the staining is similar, that would suggest that the determinant of WHAMM localization to autophagosomes and autolysosomes is membrane curvature, which is reported here. This experiment would really help connect the dots.

2 Mcherry-WHAMM is not localized when it performs its late function, which is most significant here. How does WHAMM promote tubule extension? Does WHAMM localize at the base of membrane tubules, a WASH, or along their length? The NPF activity is generated a pushing or compressive force through branched actin network. Depending on precise WHAMM localization, the model is significantly different. FigS2e does not suffice in this respect. The resolution is also not enough. Cortactin would also probably be a better marker for branched actin network than Lifeact.

3 Membrane tubules are best seen in live imaging and surprisingly no supplementary movie is provided, even though most stills come from live imaging. I believe that CK666 wash-out may provide a burst of tubules that would be very nice to image. This system may help as well the localization of mCherry-WHAMM during tubulogenesis.

Suggestions

4 Even though this study is well performed and self-consistent, it would be interesting to see if one can understand the reason for discrepancies with previous reports using simple experiments. Of course, it can simply be cell type differences. But there might also be functional redundancy with JMY, the WHAMM paralog, which has already been implicated in autophagy multiple times, and the WHAMM KO, unlike transient siRNA transfection, might give time for cells to adapt, for example, through JMY upregulation. In their NRK cell system, the authors should knock-down WHAMM, JMY or both, using siRNAs to see whether autophagy phenotypes are early or late. That would be great for once to have a study that tries to solve issues with equally well performed earlier studies.

5 A phenotype that has been seen with N-WASP, WAVE and WASH is that NPF recruitment at their respective membranes is enhanced and that their dynamics are completely frozen when actin polymerization is not permitted (Weisswange Nature 2009, Millius Curr Biol 2009, Derivery PLOS One 2012). That would be great to see if this extends to WHAMM/JMY family and would thus be a general mechanism of NPFs. The predictions are clear: The WHAMM mutants that do not polymerize actin should associate more with autolysosomes, they should not be dynamic by FRAP. When CK666 is applied, there should also be more WT WHAMM at the surface of autolysosomes and it should also be much less dynamics. These observations would be well connected to WHAMM NPF mutants and CK666 experiments. If these observations are indeed general to all NPFs, that would suggest a mechanism by which the retrograde flow of actin and of the Arp2/3 complex promotes the detachment of NPFs and hence their dynamic turn-over.

6 In the discussion section, WHAMM is compared to the role of NWASP at the clathrin coated pit. I believe that the comparison of WHAMM would be richer with the role of WASH at endosomes. WASH, unlike NWASP, also involves interaction with the microtubule system and membrane tubulation. With WASH inactivation also, endosomes grow bigger, similar to the bigger autolysosomes observed when WHAMM is inactivated.

Miscellaneous

7 tabulation (Fig.7) -> tubulation

8 What does transact really mean ? I believe that this jargon is not required.

9 FigS5. Which clathrin light chain is tagged in this experiment ? They may not be equivalent.

Reviewer #3 (Remarks to the Author):

"WHAMM initiates autolysosome tubulation by promoting actin polymerization on autolysosomes" by Anbang Dai, Li Yu, and Hong-Wei Wang

This is a significant study, and an important follow-up to high profile prior work on the question on 'what happens with lysosomes in cells after autophagy', now reporting the "how" aspect of it. I recommend revision, with one major task I would ask of the authors, and that is to compare this with tubulation of conventional lysosomes (sic, not autolysosomes) within the classical endolysosome system, if possible.

In this study, Dai et al. analyzed in detail a new role of WHAMM, a WASP family protein, in the regulation of autophagic lysosomal reformation (ALR) by promoting actin polymerization on autolysosomes. WHAMM is recruited to autolysosomes through its binding to the phospholipid PI(4,5)P2, and two amino acid motifs on the N-terminal side of WHAMM are responsible for the binding. At the autolysosomes, WHAMM acts as a nucleation factor (NPF) to assemble actin networks, which in turn promote autolysosome tubulation. This study adds a new player for the regulation of ALR, further stressing the importance of actin network dynamics in the process of lysosome reformation, in addition to its roles in autolysosome formation. The authors provide solid data in this study, with well designed experiments and clear conclusions.

Major points:

1. I would ask the authors, if possible, to compare ALR with the tubulation of conventional lysosomes (sic, not autolysosomes but just garden variety lysosomes) within the classical endolysosome system. For example, it is known that in macrophages lysosomes can form tubular networks.
2. Considering the manuscript as a whole, this study would gain more significance if the authors could provide some evidence regarding which signals recruit WHAMM to the newly formed autolysosomes. Is it simply mediated by the accumulation of PI(4,5)P2 on the autolysosomes? But why is PIP2 accumulating? It is recognized that, in the Discussion, the authors have already raised the question as to why WHAMM localizes at the ALR membranous region but not the plasma membrane. There may be a clue in there. However, whether this needs to be experientially addressed here I'd leave up to the authors and editors.

There are several moderate to minor issues in the manuscript:

1. Where is WHAMM localized before autophagy induction?
2. In Fig. 1b, the images for KO cells in the lower panel are apparently brighter than the WT cells in the upper panel; they need to be adjusted to equal levels.
3. In Fig. 1d, why do the LC3 bands have a pattern with the upper band disappearing after starvation while the lower band shows almost no change?
4. In quantification figures with percentages, e.g. Fig. 1g, Fig. 2b, Fig. 3c, 3f and others, it seems that 0.1 is probably 10%, and 0.2 is 20%; so, it's better if the authors can make it clearer in graphs with the Y axis labels.
5. How exactly to define tubular autolysosomes; and how to determine autolysosome size? Did the authors make it clear in Methods? It was not possible to find this key information. A detailed methodology description of this important methodology is needed to buttress the authenticity and quantification/reproducibility.
6. Fig. 3b should include the staining of FL or mutant WHAMM proteins expressed in the WHAMM-KO cells in order to show its colocalization with mCherry-LAMP1; it's important to show the location of

WHAMM relative to tubular autolysosome structures.

7. In the text, Line 190, the authors wrote that "amino acids 1-340 are required for binding to PI(4,5)P2 (Fig. 5a)". However, Fig. 5a shows that amino acids 261-809 and amino acids 310-809 can both bind to PI(4,5)P2 efficiently. Further analysis in Fig. 5b indicates that there are two PI(4,5)P2-binding motifs on WHAMM, one before amino acid 260, the other one after amino acid 310. So, the text on Line 190 needs careful statements and clarification.

8. In the text, Line 200, the authors should include detailed reasons why the two amino acid regions, 188-208 and 319-339 of WHAMM were selected as potential PI(4,5)P2-binding motifs.

9. For Fig. 6, the authors should use suitable and authenticated probes to show PIP2 colocalization with WT or mutant WHAMM in the cells after autolysosome induction.

10. There are a few writing errors in the manuscript. For example, Line 69, better to use "monitored" or "measured" rather than "checked"; Line 127, "regulates" shouldn't have "s"; Line 143, "we" can be changed to "and".

Dear Editor and Reviewers,

We have carried out extensive additional experimental work to address the queries of the reviewers. Please find below our point-by-point discussion in which we reiterate the reviewers' points and provide answers. Our responses are in italic font.

Reviewer #1 (Remarks to the Author):

This study focuses on the role of WHAMM in lysosome reformation from autolysosomes. The authors demonstrate that the protein is required for not autophagosome formation but for lysosome reformation in contrast to the previous work from the other group. They show that WHAMM promotes actin network formation on autolysosomes through its actin nucleation ability, which is involved in autolysosome tubulation, an initial step of lysosome reformation. They further found that WHAMM is recruited to autolysosomes by PI(4,5)P2 and its binding to this PIP is essential for lysosome reformation. They locate the PIP binding site in WHAMM.

Overall this paper is logical and well written. The data is timely and provides new mechanistic insights into lysosome reformation from autolysosomes, which is an important step in autophagy but still not well understood. I expect that this work will be not only of interest to the autophagy community but also of broad interest to researchers in membrane biology field, providing a conceptual advance for understanding how cells manage organelle dynamics.

Needed improvements:

1. To confirm that the structures in TEM images in Fig 2e are identical to dots labeled with both LC3 and LAMP1 in Fig 2c, the authors should perform CLEM (Correlative light and electron microscopy) method or immuno EM. In Fig 2e, vesicles in the KO cells include a lot of membranes, suggesting that degradation inside autolysosomes is suppressed. The authors should examine a possibility that WHAMM is also involved in autolysosomal degradation by autophagy flux assay.

Response: We thank the reviewer for this insightful comment. As the reviewer suggested, we performed the CLEM experiments for both WT and KO cells (Fig.

R1a). We found that the LC3- and LAMP1-positive structures are indeed autolysosomes. This data is now updated in the revised manuscript as Fig. 2g.

As the reviewer suggested, we carried out autophagy flux assays. We found that both WT and WHAMM-KO cells showed similar accumulation of LC3-II after BafA treatment (Fig. R1b). This suggests that the autolysosomal degradation is probably not affected in WHAMM-KO cells. This data is now updated in the revised manuscript to replace previous data as Fig. 1d.

Figure R1 (a) Correlative Light Electron Microscopy of WT and WHAMM-KO cells after starvation for 12 hours. Both WT and WHAMM-KO NRK cells were transfected with CFP-LC3 (green) and LAMP-mCherry (red). 18 hours post transfection, cells were starved for 12 hours and then fixed with 4% paraformaldehyde (PFA). Fluorescence images were taken using a confocal microscope. Cells were then further fixed with 2.5% glutaraldehyde (GA) and prepared for EM analysis. A detailed description can be found in the Methods section of the revised manuscript. (Scale bar, main figure 5 μ m; right panels 500 nm). (b) Autophagic flux was monitored by treating both WT and KO cells with Bafilomycin-A1 (BafA) after starvation.

2. In WHAMM-KO cells, the actin-positive autolysosomes are reduced (Fig 3h, i). The authors should show that FL but not 1-630 and W807A mutants of WHAMM rescue the phenotype.

Response: We thank the reviewer for this insightful comment. As the reviewer suggested, we carried out the rescue assay in WHAMM-KO cells using FL WHAMM and the two NPF mutants (1-630, W807A). We found that FL partially rescued the recruitment of actin to autolysosomes while the NPF mutants did not (Fig. R2a, R2b). This result further shows that WHAMM directs the formation of actin networks on autolysosomes. This data is now included in the revised manuscript as Supplementary Fig. 4b. Statistics result is updated in the revised manuscript as Fig. 3h.

Figure R2 The NPF activity of WHAMM is required for F-actin localization on autolysosomes. (a) WHAMM-KO cells were transfected with GFP-LifeAct and WHAMM-FL or different NPF-defective mutants. 18 hours post transfection, cells were starved for 8 hours and fixed and stained with antibody against LAMP1. (Scale bar, main figure 5 μ m; upper right panel, 2 μ m). (b) Cells in (a) were assessed for actin-positive LAMP1 structures and compared with cells in Fig 3g. n=26 (WT), 25 (KO), 22 (FL), 26 (1-630), 21 (W807A) from two independent experiments. Error bars indicate SEM. Statistical analysis was performed using one-way ANOVA followed

*with Holm-Sidak's multiple comparisons test. Compared with WT: ###, $p < 0.001$; #, $p < 0.05$; ns, not significant. Compared with KO: *, $p < 0.05$, ns, not significant.*

3. As the authors discuss, since the plasma membrane contains a lot of PI(4,5)P₂, there must be other factor(s) that determine the specific binding of WHAMM to autolysosomes. Although the authors mention that future studies may reveal this point, it is desirable that the present paper includes identification of a candidate of binding partner of WHAMM on autolysosomes other than PI(4,5)P₂.

Response: We thank the reviewer for this important suggestion. We agree that there must be other factors that determine the specific localization of WHAMM to autolysosomes and we are very keen to identify them in our future studies. However, given the amount of work required, we feel that the identification and characterization of these potential novel factors is beyond the scope of the current study, which focuses mainly on the roles of actin and WHAMM in regulation of ALR.

Reviewer #2 (Remarks to the Author):

In this manuscript, Dai and colleagues report a novel function of WHAMM in autophagic lysosome reformation, which is a late step of autophagy. To this end, they have generated WHAMM KO in NRK cells and they perform rescue with different constructs of WHAMM. In particular, they show using specific mutants that Arp2/3 activation and PIP₂ binding are critical for WHAMM to perform its late function in autophagy.

Overall this manuscript is nice and clear. It is well written and the reader has a sense of logical progression until the end. The techniques are appropriate and the results are convincing. In terms of originality, there were already reports that WHAMM was involved in autophagy, but only in an early step, autophagosome biogenesis. The implication of WHAMM in late autophagy as described here is completely novel and well documented in the manuscript. A conundrum though is that the early function was revealed using siRNAs, whereas the late function described here is revealed using CRISPR-Cas9 generated KO, so incomplete knock-down is not the reason...

I believe that this work should be published provided that minor improvements are included. I also suggest experiments that may extend the manuscript. As such, they are not absolute requirements, but to my sense would increase the number of reasons of citing this manuscript when it is going to be published.

Required improvements

1 The actin mutants of WHAMM were previously described; the PIP2 mutants are new and are clearly a major advance of the manuscript. The PIP2 mutants are inactive as expected. However, I believe that a clear co-localization of WT WHAMM with a PIP2 probe is lacking. To most investigators, PIP2 is mostly located at the plasma membrane. Here it would also be useful to see the relative staining of autolysosomes and plasma membrane. If the staining is similar, that would suggest that the determinant of WHAMM localization to autophagosomes and autolysosomes is membrane curvature, which is reported here. This experiment would really help connect the dots.

Response: We thank the reviewer for this important suggestion. As demonstrated in our previous studies, PI(4,5)P₂ plays a central role in recruiting and regulating the recruitment of Clathrin machinery and kinesin during ALR^{1, 2}. Also, the PI(4,5)P₂ on autolysosomes is mainly generated by PIP5K1B¹. We introduced mCherry-PLCδ-PH as a marker to label cellular PI(4,5)P₂. Before starvation was induced, mCherry-PLCδ-PH showed few cellular puncta. But after starvation, we observed an increased intracellular pool of PI(4,5)P₂ (Fig. R3a). We also observed co-localization or juxtaposition of WHAMM with mCherry-PLCδ-PH on autolysosomes under starvation (Fig. R3b). Moreover, PIP5K1B is also co-localized with WHAMM (Fig. R3c). Fig. R3b is now included in the revised manuscript as Fig. 7a.

Figure R3 WHAMM co-localizes with PI(4,5)P₂ probes on autolysosomes. (a) WT NRK cells were transfected with mCherry-PLCδPH. 18 hours post transfection, cells were starved for up to 8 hours and observed using live-cell imaging. (b) WT NRK cells stably expressing GFP-WHAMM were transfected with mCherry-PLCδPH. 18 hours post transfection, cells were starved for 4 hours in the presence of 100 nM Lysotracker Deep Red and observed using live-cell imaging. Dashed squares showed WHAMM's localization correlates to Plasma membrane (a) and autolysosomes (b). (scale bar, main figure 5 μm; right panel 1 μm). (c) WT NRK cells stably expressing mCherry-WHAMM were transfected with GFP-PIP5K1B. 18 hours post transfection, cells were starved for 4 hours, then fixed and stained with antibody against LAMP1. Arrowheads indicate co-localized WHAMM and PIP5K1B on autolysosomes. (Scale bar, main figure 5 μm; upper right panel, 2 μm). (b) and (c) in this figure were deconvolved using the built-in software (NIS-elements, Nikon).

2 Mcherry-WHAMM is not localized when it performs its late function, which most significant here. How does WHAMM promote tubule extension ? Does WHAMM localize at the base of membrane tubules, a la WASH, or along their length ? The NPF activity is generated a pushing or compressive force through branched actin network. Depending on precise WHAMM localization, the model is significantly different. FigS2e does not suffice in this respect. The resolution is also not enough. Cortactin would also probably be a better marker for branched actin network than Lifeact.

Response: We thank the reviewer for this insightful comment. To show WHAMM's localization relative to the reformation tubule more precisely, we generated a stable cell line expressing GFP-WHAMM. We then applied live-cell imaging and found that WHAMM puncta localized on the surface of autolysosomes (Fig. R4a, Supplementary Movie 1) and at the base of a newly formed reformation tubule (Fig. R4b, Supplementary Movie 2). Moreover, although WHAMN puncta occasionally appeared on the reformation tubules (Fig. R4c), WHAMM did not fully cover the surface of reformation tubules.

As the reviewer suggested, we tested other markers for the branched actin network. We used mCherry-Arp3 and mCherry-Cortactin to label the branched actin network. Indeed, we observed that WHAMM co-localizes with these markers on autolysosomes (Fig. R4d, R4e). We thus conclude that WHAMM generates branched actin networks on autolysosomes.

Taken together, these data offer direct evidence for our proposed model that WHAMM promotes branched actin network formation to facilitate autolysosome tubulation from the base of the reformation tubule. These new data is now included in the revised manuscript as Fig. 4.

Figure R4 WHAMM generates branched actin networks on autolysosomes and the base of reformation tubule from autolysosomes. (a) WT NRK cells stably expressing GFP-WHAMM were transfected with LAMP1-mCherry, 18 hours post-transfection, cells were starved for 8 hours and observed using live-cell imaging. GFP-WHAMM was pseudo-colored to cyan and LAMP1-mCherry was pseudo-colored to yellow.

Arrowheads indicate WHAMM puncta on the surface of autolysosomes. (Scale bar, main figure 5 μm ; upper right panel, 2 μm). (b) Time-lapse images were taken of cells in (a). Arrows indicate WHAMM puncta at the neck of a newly-formed reformation tubule from an autolysosome. (Scale bar, 2 μm). (c) A snapshot of an elongated reformation tubule was extracted from cells in (a). Arrows indicate WHAMM puncta on the reformation tubule. (d-e) WT NRK cells stably expressing GFP-WHAMM were transfected with mCherry-Arp3 and mCherry-Cortactin, respectively. 18 hours post transfection, cells were starved for 4 hours and then fixed and stained with antibody against LAMP1. Arrowheads indicate co-localized WHAMM-Arp3 (d) or WHAMM-Cortactin (e) on autolysosomes. (Scale bar, main figure 5 μm ; upper right panel, 2 μm). All images in this figure were deconvolved using the built-in software (NIS-elements, Nikon).

3 Membrane tubules are best seen in live imaging and surprisingly no supplementary movie is provided, even though most stills come from live imaging. I believe that CK666 wash-out may provide a burst of tubules that would be very nice to image. This system may help as well the localization of mCherry-WHAMM during tubulogenesis

Response: We thank the reviewer for this great idea. As the reviewer suggested, we carried out the CK666 wash-out experiment. Indeed, we did observe a burst of tubule formation (Fig. R5, Supplementary Movie 3). Even though we believe our data in Figure R4 is sufficient to reveal WHAMM's localization during tubulogenesis, we really appreciate this great idea from the reviewer. It would be really helpful for future studies on ALR. This data is now included in the revised manuscript as Supplementary Fig. 3.

Figure R5 Recovery of tubulation events after CK666 wash-out. Scale bar, 5 μm .

Suggestions

4 Even though this study is well performed and self-consistent, it would be interesting to see if one can understand the reason for discrepancies with previous reports using simple experiments. Of course, it can simply be cell type differences. But there might also be functional redundancy with JMY, the WHAMM paralog, which has already been implicated in autophagy multiple times, and the WHAMM KO, unlike transient siRNA transfection, might give time for cells to adapt, for example, through JMY upregulation. In their NRK cell system, the authors should knock-down WHAMM, JMY or both, using siRNAs to see whether autophagy phenotypes are early or late. That would be great for once to have a study that tries to solve issues with equally well performed earlier studies.

Response: We thank the reviewer for this constructive suggestion. As the reviewer suggested, we knocked down JMY in WHAMM KO cells, and measured autophagy by three different assays (LC3-II conversion, p62 degradation and LC3 punctum formation). We were surprised that knockdown of JMY only had a marginal effect on autophagy in the WHAMM-KO cells that we are using (Fig. R6). We repeated this experiment multiple times and confirmed that the result we show here is reproducible. From these data, we cannot attribute the discrepancy with previous publications to a compensation effect of JMY.

Figure R6 WHAMM-KO cells with JMY knockdown show a normal level of autophagy activity. (a) WHAMM-KO cells were transfected with non-specific (NC) or JMY siRNA. 48 hours after initial transfection, cells were transfected with another round of siRNA and further cultured for 48 hours (total = 96 hours). Then cells were starved for 4 hours with or without Bafilomycin-A1 (BafA). After starvation, cells were collected and analyzed by western blot with antibodies against p62, LC3 and actin. (b) The LC3-II level in cells in (a) treated with BafA was first normalized with actin and the relative LC3-II level in each RNAi assay was normalized with NC. (c) The p62 level was first normalized with actin and the relative p62 level in each RNAi assay was normalized with NC before starvation. (d) Cells in (a) without BafA treatment were fixed and stained with antibodies against LC3 after 2 hours of starvation. (e) LC3-positive structures in cells from (d) were counted and plotted. n=45 (NC), 49 (siJMY). Two-tailed t test. ns, not significant. (f) JMY's mRNA levels in the above experiments were measured using qPCR. Band intensity was measured using ImageJ. All LC3-II turn-over and p62 degradation experiments were performed 3 times.

5 A phenotype that has been seen with N-WASP, WAVE and WASH is that NPF recruitment at their respective membranes is enhanced and that their dynamics are completely frozen when actin polymerization is not permitted (Weisswange Nature 2009, Millius Curr Biol 2009, Derivery PLOS One 2012). That would be great to see if this extends to WHAMM/JMY family and would thus be a general mechanism of NPFs. The predictions are clear: The WHAMM mutants that do not polymerize actin should associate more with autolysosomes, they should not be dynamic by FRAP. When CK666 is applied, there should also be more WT WHAMM at the surface of autolysosomes and it should also be much less dynamics. These observations would be well connected to WHAMM NPF mutants and CK666 experiments. If these observations are indeed general to all NPFs, that would suggest a mechanism by which the retrograde flow of actin and of the Arp2/3 complex promotes the detachment of NPFs and hence their dynamic turn-over.

Response: We thank the reviewer for this helpful idea. It would be great to investigate this general mechanism regarding the turn-over of NPFs. However, given that the

main focus in this study to identify WHAMM's general roles in regulation of ALR, we thought it would be more appropriate to perform these experiments in an independent study, which will allow us to carry out a thorough investigation using both in vitro and in vivo experiments.

6 In the discussion section, WHAMM is compared to the role of NWASP at the clathrin coated pit. I believe that the comparison of WHAMM would be richer with the role of WASH at endosomes. WASH, unlike NWASP, also involves interaction with the microtubule system and membrane tubulation. With WASH inactivation also, endosomes grow bigger, similar to the bigger autolysosomes observed when WHAMM is inactivated.

Response: We thank the reviewer for this great idea! As the reviewer suggested, we have now compared WHAMM's role in ALR with WASH on endosomes in the discussion section of the revised manuscript at line 300-305 (highlighted by yellow in the revised main text).

Miscellaneous

7 tabulation (Fig.7) -> tubulation

8 What does transact really mean? I believe that this jargon is not required.

9 FigS5. Which clathrin light chain is tagged in this experiment ? They may not be equivalent.

Response: We thank the reviewer for pointing out these miscellaneous issues. We have made the corrections accordingly.

Reviewer #3 (Remarks to the Author):

“WHAMM initiates autolysosome tubulation by promoting actin polymerization on autolysosomes” by Anbang Dai, Li Yu, and Hong-Wei Wang

This is a significant study, and an important follow-up to high profile prior work on the question on 'what happens with lysosomes in cells after autophagy', now reporting the "how" aspect of it. I recommend revision, with one major task I would ask of the authors, and that is to compare this with tubulation of conventional lysosomes (sic,

not autolysosomes) within the classical endolysosome system, if possible.

In this study, Dai et al. analyzed in detail a new role of WHAMM, a WASP family protein, in the regulation of autophagic lysosomal reformation (ALR) by promoting actin polymerization on autolysosomes. WHAMM is recruited to autolysosomes through its binding to the phospholipid PI(4,5)P₂, and two amino acid motifs on the N-terminal side of WHAMM are responsible for the binding. At the autolysosomes, WHAMM acts as a nucleation factor (NPF) to assemble actin networks, which in turn promote autolysosome tubulation. This study adds a new player for the regulation of ALR, further stressing the importance of actin network dynamics in the process of lysosome reformation, in addition to its roles in autolysosome formation. The authors provide solid data in this study, with well designed experiments and clear conclusions.

Major points:

1. I would ask the authors, if possible, to compare ALR with the tubulation of conventional lysosomes (sic, not autolysosomes but just garden variety lysosomes) within the classical endolysosome system. For example, it is known that in macrophages lysosomes can form tubular networks.

Response: We thank the reviewer for this important suggestion. Lysosome tubulation has been observed in LPS-exposed macrophages and dendritic cells³. Although ALR and lysosome tubulation are induced by different stimuli, the similarities between these two processes in macrophages and dendritic cells are striking. Both are dependent on microtubules and driven by microtubule-based motors²⁻⁵, and both are governed by mTOR and regulated by Rab7⁶. Given so many similarities between ALR and lysosome tubulation, we speculate that they share the same regulatory pathway. We have put this in the discussion of the revised manuscript at line 306-312. (highlighted by yellow in the revised main text).

2. Considering the manuscript as a whole, this study would gain more significance if the authors could provide some evidence regarding which signals recruit WHAMM to the newly formed autolysosomes. Is it simply mediated by the accumulation of PI(4,5)P₂ on the autolysosomes? But why is PIP₂ accumulating? It is recognized that,

in the Discussion, the authors have already raised the question as to why WHAMM localizes at the ALR membranous region but not the plasma membrane. There may be a clue in there. However, whether this needs to be experientially addressed here I'd leave up to the authors and editors.

Response: We thank the reviewer for this great suggestion. We agree that there must be other factors that determine the specific localization of WHAMM to autolysosomes and we are very keen to identify them in our future studies. However, given the amount of work required, we feel that the identification and characterization of these potential novel factors is beyond the scope of the current study, which focuses mainly on the roles of actin and WHAMM in regulation of ALR.

As for the accumulation of PI(4,5)P₂ on autolysosomes, we showed in our previous studies that PI(4,5)P₂ is locally generated by PIP5K1B¹, which is recruited to the surface of autolysosomes during ALR.

There are several moderate to minor issues in the manuscript:

1. Where is WHAMM localized before autophagy induction?

Response: We thank the reviewer for this important question. Previous reports have shown that WHAMM primarily associates with the ER, cis-Golgi network and DFCEP1 under fed conditions⁷⁻⁹. Following the reviewer's suggestion, we examined WHAMM's localization in relation to some cellular compartments before autophagy induction in our NRK cells. We found that WHAMM did not co-localize with the mitochondrion marker Tomm20, the early endosome marker Rab5, or the recycling endosome marker Rab11. However, WHAMM did co-localize with the ER marker Sec61β (Fig. R8a), in agreement with previous reports⁸, and with the late endosome marker Rab7. WHAMM's association with Rab7 raises the possibility that WHAMM might also be delivered from endosomes to lysosomes. This data is now included in the revised manuscript as Supplementary Fig. 8.

Figure R8 WHAMM's localization before autophagy induction. (a) Constructs expressing markers for each designated cellular compartment were transfected in cells stably expressing GFP-WHAMM or mCherry-WHAMM. 18 hours post-transfection, cells were observed under a confocal microscope. (Scale bar, main figure 5 μm ; right panel, 2 μm)

2. In Fig. 1b, the images for KO cells in the lower panel are apparently brighter than the WT cells in the upper panel; they need to be adjusted to equal levels.

Response: We thank the reviewer for this careful inspection. The difference in brightness is due to the different expression level of LC3. We have chosen cells with similar expression levels in the revised manuscript.

3. In Fig. 1d, why do the LC3 bands have a pattern with the upper band disappearing after starvation while the lower band shows almost no change?

Response: We thank the reviewer for this question. We think that it's probably due to a high basal level of autophagy causing a thicker LC3-II band even before the starvation. To better address this problem, we redid this experiment by also introducing Bafilomycin-A1 to examine the autophagic flux. As shown in Fig. R9, LC3-II showed similar accumulation in both WT and KO cells after Bafilomycin-A1 treatment, which suggests that the autophagy activity is normal in KO cells. We have updated the manuscript with this new figure panel as Fig. 1d.

Figure R9 Autophagic activity was monitored by treating WT and KO cells with Bafilomycin-A1 (BafA) after inducing starvation.

4. In quantification figures with percentages, e.g. Fig. 1g, Fig. 2b, Fig. 3c, 3f and others, it seems that 0.1 is probably 10%, and 0.2 is 20%; so, it's better if the authors can make it clearer in graphs with the Y axis labels.

Response: Point taken. We changed the Y-axis labels in the revised manuscript.

5. How exactly to define tubular autolysosomes; and how to determine autolysosome size? Did the authors make it clear in Methods? It was not possible to find this key

information. A detailed methodology description of this important methodology is needed to buttress the authenticity and quantification/reproducibility.

Response: We thank the reviewer for pointing out this issue. A detailed description of the methodology is presented below and has also been added to the Methods section of revised manuscript at line 403-408 (highlighted by yellow in the revised main text).

“Identification and measurement of autolysosomes and tubular structures. Tubular autolysosomes are LAMP1 positive tubular structures extending from LAMP1-LC3 positive vesicular autolysosomes. This feature allows us to distinguish between autolysosomes and lysosomes when measuring the size of autolysosomes. The size of autolysosomes was measured using Image by encircling the autolysosomes using the oval selection tool. The area was then measured accordingly.”

6. Fig. 3b should include the staining of FL or mutant WHAMM proteins expressed in the WHAMM-KO cells in order to show its co-localization with mCherry-LAMP1; it's important to show the location of WHAMM relative to tubular autolysosome structures.

Response: We thank the reviewer for this important suggestion. We repeated this experiment as shown in figure R10 with better representative images. FL and mutant WHAMM are localized on the main body of autolysosomes. There is very little WHAMM signal on the reformation tubules. This data is now updated in the revised manuscript as Fig. 3b and Supplementary Fig. 2a.

Figure R10 Staining of FL and mutant WHAMM during ALR (In relation to Fig. 3b and Supplementary Fig. 2a). WHAMM FL or NPF-defective mutants were transfected into WHAMM-KO cells stably expressing LAMP1-YFP. 18 hours post-transfection, cells were starved for 8 hours, then observed by confocal microscopy. WHAMM constructs were pseudo-colored to cyan. Scale bar, 5 μ m

7. In the text, Line 190, the authors wrote that “amino acids 1-340 are required for binding to PI(4,5)P2 (Fig. 5a)”. However, Fig. 5a shows that amino acids 261-809 and amino acids 310-809 can both bind to PI(4,5)P2 efficiently. Further analysis in Fig. 5b indicates that there are two PI(4,5)P2-binding motifs on WHAMM, one before amino acid 260, the other one after amino acid 310. So, the text on Line 190 needs careful statements and clarification.

Response: Point taken. We fixed this issue in the revised manuscript at line 217-219.

8. In the text, Line 200, the authors should include detailed reasons why the two

amino acid regions, 188-208 and 319-339 of WHAMM were selected as potential PI(4,5)P₂-binding motifs.

Response: We appreciate the reviewer's careful inspection. The reasons why we consider these two regions as candidates are presented below and also updated in the revised manuscript at line 225-236 (highlighted by yellow in the revised main text).

“We closely inspected the N-terminal region of WHAMM for potential phospholipid binding motifs and found two interesting helices: helix 1 (188-208) and helix 2 (319-339), which are located separately within residues 1-260 and 260-340 (Fig. 6b, 6c). These two helices are predicted by secondary structure algorithms and are conserved among species (Fig. 6c and Supplementary Fig. 5a). Using helical wheel representations, we found that both helices have similar amphipathicity and residue arrangements, with basic residues on both sides of the hydrophobic face, which could potentially bind membranes comprising lipids with negatively charged head groups (Fig. 6d). Helices with such properties have also been found in other membrane-binding proteins such as ATG3¹⁰ and BAR domain-containing proteins¹¹, and the helices are responsible for the interaction of the proteins with negatively charged lipids. Taking these features together, we suspected that these two regions might be responsible for the interaction of WHAMM with membranes containing PI(4,5)P₂.”

9. For Fig. 6, the authors should use suitable and authenticated probes to show PIP₂ co-localization with WT or mutant WHAMM in the cells after autolysosome induction.”

Response: We thank the reviewer for this important suggestion. We did this experiment and the results are shown in Figure R3.

Figure R3 WHAMM co-localizes with PI(4,5)P₂ probes on autolysosomes. (a) WT NRK cells were transfected with mCherry-PLCδPH. 18 hours post transfection, cells were starved for up to 8 hours and observed using live-cell imaging. (b) WT NRK cells stably expressing GFP-WHAMM were transfected with mCherry-PLCδPH. 18 hours post transfection, cells were starved for 4 hours in the presence of 100 nM Lysotracker Deep Red and observed using live-cell imaging. Dashed squares showed WHAMM's localization corresponding to Plasma membrane (a) and autolysosomes (b). (c) WT NRK cells stably expressing mCherry-WHAMM were transfected with GFP-PIP5K1B. 18 hours post transfection, cells were starved for 4 hours, then fixed and stained with antibody against LAMP1. Arrowheads indicate co-localized WHAMM and PIP5K1B on autolysosomes. (Scale bar, main figure 5 μm; upper right panel, 2 μm). (b) and (c) in this figure were deconvolved using the built-in software (NIS-elements, Nikon).

10. There are a few writing errors in the manuscript. For example, Line 69, better to use “monitored” or “measured” rather than “checked”; Line 127, “regulates” shouldn’t have “s”; Line 143, “we” can be changed to “and”.

Response: We thank the reviewer for this careful inspection. These writing errors have been corrected according to the reviewer’s suggestions.

References:

1. Rong, Y. *et al.* Clathrin and phosphatidylinositol-4,5-bisphosphate regulate autophagic lysosome reformation. *Nature Cell Biology* **14**, 924-934 (2012).
2. Du, W. *et al.* Kinesin 1 Drives Autolysosome Tubulation. *Developmental Cell* **37**, 326-336 (2016).
3. Saric, A. *et al.* mTOR controls lysosome tubulation and antigen presentation in macrophages and dendritic cells. *Molecular Biology of the Cell* **27**, 321-333 (2016).
4. Harrison, R.E., Bucci, C., Vieira, O.V., Schroer, T.A. & Grinstein, S. Phagosomes fuse with late endosomes and/or lysosomes by extension of membrane protrusions along microtubules: role of Rab7 and RILP. *Molecular and cellular biology* **23**, 6494-6506 (2003).
5. Li, X. *et al.* A molecular mechanism to regulate lysosome motility for lysosome positioning and tubulation. *Nature Cell Biology* **18**, 404-417 (2016).
6. Yu, L. *et al.* Termination of autophagy and reformation of lysosomes regulated by mTOR. *Nature* **465**, 942-946 (2010).
7. Campellone, K.G., Webb, N.J., Znameroski, E.A. & Welch, M.D. WHAMM Is an Arp2/3 Complex Activator That Binds Microtubules and Functions in ER to Golgi Transport. *Cell* **134**, 148-161 (2008).
8. Kast, David J., Zajac, Allison L., Holzbaur, Erika L.F., Ostap, E.M. & Dominguez, R. WHAMM Directs the Arp2/3 Complex to the ER for Autophagosome Biogenesis through an Actin Comet Tail Mechanism. *Current Biology* **25**, 1791-1797 (2015).
9. Mathiowetz, A.J. *et al.* An Amish founder mutation disrupts a PI(3)P-WHAMM-Arp2/3 complex-driven autophagosomal remodeling pathway.

Molecular Biology of the Cell **28**, 2492-2507 (2017).

10. Nath, S. *et al.* Lipidation of the LC3/GABARAP family of autophagy proteins relies on a membrane-curvature-sensing domain in Atg3. *Nature Cell Biology* **16**, 415-424 (2014).
11. Bhatia, V.K. *et al.* Amphipathic motifs in BAR domains are essential for membrane curvature sensing. *The EMBO Journal* **28**, 3303-3314 (2009).

REVIEWERS' COMMENTS:

Reviewer #1 (Remarks to the Author):

I am satisfied with the authors' response to my requests.

Reviewer #2 (Remarks to the Author):

The revised manuscript by Dai et al. satisfactorily contains the improvements that I had suggested. I now recommend publication of the manuscript in its current form.

Alexis Gautreau

Reviewer #3 (Remarks to the Author):

This is well revised and recommended for publication